# *COMPASS*: A Multi-Turn Benchmark for Tool-Mediated Planning & Preference Optimization

## Abstract

Real-world tasks like travel planning require LLM agents to satisfy hard constraints (dates, budget) while optimizing user's utility preferences (cheapest hotel, most convenient flights). We formalize this as *constrained preference optimization*, where agents strategically use tools to gather information and compare options to optimize user's preferences. We introduce **COMPASS** (**C**onstrained **O**ptimization through **M**ulti-turn **P**lanning **a**nd **S**trategic **S**olutions), a benchmark evaluating agents through realistic travel planning. We build a travel database covering transportation, accommodation, and ticketing for 20 U.S. National Parks, plus a tool ecosystem mirroring commercial booking platforms. Evaluating state-of-the-art models reveals a significant **acceptable–optimal gap**: models achieve 85-95% constraint satisfaction but only 60-70% preference optimization, settling for feasible rather than optimal solutions. Performance degrades sharply on multi-service coordination tasks. Our tool-use analysis shows task success strongly correlates with information gathering—insufficient exploration is the primary bottleneck, though future models should prioritize efficient over exhaustive search. *COMPASS* provides a rigorous benchmark for diagnosing core challenges in constrained preference optimization and guiding development of user-aligned agents.

## 1 Introduction

As Large Language Models (LLMs) move toward real-world deployment, their role is increasingly to assist users with everyday tasks: travel planning, shopping, scheduling. These tasks inherently involve trade-offs between requirements and preferences. When planning a trip, users have hard requirements they cannot compromise on: specific dates, budget limits. But within these constraints, they want to maximize their utility: find the cheapest hotel, book the most convenient flights, or select the highest-rated accommodations. This pattern of decision-making under hard constraints while optimizing for soft preferences is ubiquitous in everyday life.

We formalize this setting as *constrained preference optimization*: agents must satisfy hard constraints while optimizing soft preferences. Achieving this requires tool-mediated problem solving, where agents gather information by strategically invoking external resources to discover and compare options. The combination of constrained satisfaction and tool calling is essential for real-world deployment.

Yet current benchmarks evaluate only whether agents can find solutions that meet basic requirements—not whether they find the *best* solutions. Most planning benchmarks reduce tasks to single-turn constraint satisfaction (Xie et al., 2024; Kohli et al., 2024), where finding any feasible solution suffices. Preference optimization requires *qualitatively* different skills: agents must compare alternatives and select options that best align with user preferences. Existing tool-use benchmarks focus narrowly on whether agents call correct tools with valid parameters (Zhong et al., 2025; Yao et al., 2024), without testing adaptation to multi-turn interactions where users add requirements and provide feedback (Wang et al., 2023; Laban et al., 2025). Finally, most benchmarks rely on simplified mock databases, though database realism directly shapes task difficulty—authentic evaluation requires rich, complex environments.

To address these gaps, we introduce *COMPASS*, a benchmark instantiating constrained preference optimization through realistic travel planning. Agents must satisfy hard constraints (budget, occupancy, scheduling) while optimizing soft preferences (minimizing cost, maximizing amenities) (Fig.1 B). We construct a full environment with *(i)* a modular LLM-based user simulator for controllable

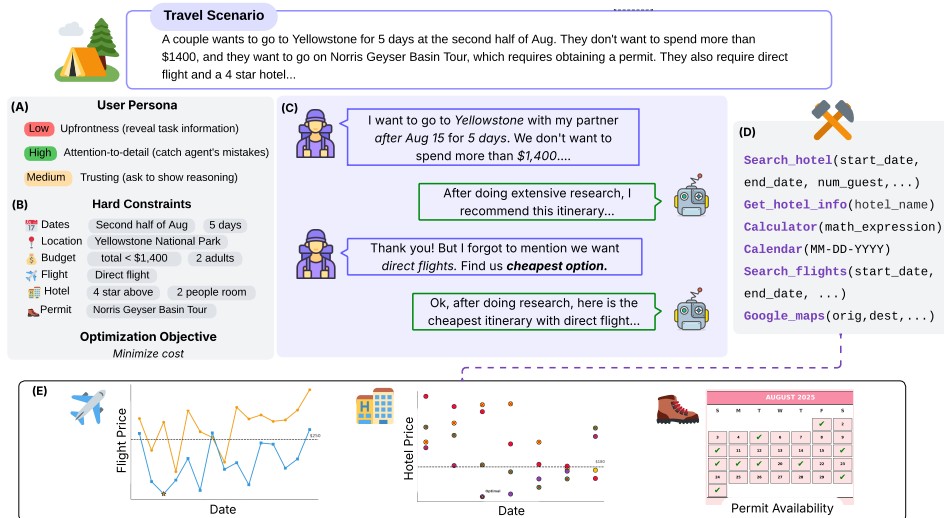

Figure 1: ***COMPASS* benchmark framework.** The environment integrates three key components for realistic evaluation of agentic capabilities in travel planning. **(A)** A modular LLM-based user simulator enables controllable multi-turn interactions, progressive constraint revelation, and diverse user personas. **(B)** We formalize travel planning as *constrained preference optimization*, where agents must satisfy hard constraints (feasible solutions) while optimizing soft user objectives (e.g., minimizing cost, maximizing amenities). **(C)** Agents interact with realistic travel databases **(E)** using a comprehensive tool ecosystem **(D)**, requiring iterative planning and refinement across conversation turns to construct optimal itineraries.

multi-turn interactions (Fig.1 *A, C*), *(ii)* a realistic database of hotels, flights, and permits from commercial sources (Fig.1 *E*), and *(iii)* a tool ecosystem mirroring booking platforms (Fig.1 *D*). Tasks are organized by **optimization types** (single metric vs. checklist maximization) and **difficulty levels** (hotel-only to full itineraries with flights and permits), providing rigorous evaluation of whether agents can deliver optimal, preference-aligned solutions.

Through extensive evaluation of state-of-the-art models, we uncover critical findings revealing fundamental limitations in current agentic capabilities. We observe a significant **acceptable-optimal gap** (Fig. 2): models achieve 85-95% constraint satisfaction but only 60-70% preference optimization, indicating agents settle for feasible solutions rather than optimizing user preferences. Performance degrades sharply as tasks require multi-service coordination (Fig. 3 *A*). While closed-source models show gradual decline, open-source models struggle dramatically with cross-component dependencies and temporal constraints—highlighting fundamental limitations in multi-step planning.

Our tool-call analysis (Sec. 6.1) reveals why models fail through three dimensions: correctness, extensiveness, and information gathering. Closed-source models invoke tools nearly flawlessly while open-source models hallucinate parameters frequently (up to 12%). Stronger models like GPT-5 make extensive tool calls (21+) without degradation, while weaker models avoid deep exploration or degrade under extended reasoning. Crucially, task success strongly correlates with information quantity—models discovering more unique information are substantially more likely to find optimal solutions, revealing **insufficient exploration** as the primary bottleneck. However, this indicates models remain in a "more search equals better performance" regime. Future models should transition toward efficient, targeted exploration—achieving high success while minimizing test-time compute.

Together, *COMPASS* serves as a rigorous benchmark for diagnosing core challenges in constrained preference optimization and guiding development of user-aligned AI agents.

## 2 RELATED WORK

Recent benchmarks have evaluated language model agents for tool use and planning, though typically in isolation. Tool-use suites such as Tau-Bench (Yao et al., 2024; Barres et al., 2025) provide

rich tool APIs and simulated user interaction, but their tasks require little long-horizon reasoning or planning. Patil et al. (2025) constructs a multi-turn user benchmark through predetermined trajectories, while others focus on coding tasks with oracle-like users (Wang et al., 2023). Planning benchmarks, in contrast, assess constraint satisfaction without interactive tool integration: Zheng et al. (2024) studies natural language planning stripped of tool complexity, and Kohli et al. (2024) frames flight booking as multiple-choice questions that bypass open-ended reasoning. Broader planning evaluations include multi-day travel planning under hard constraints (Xie et al., 2024) and frameworks for reasoning about state transitions and action consequences (Valmeekam et al., 2022). Recent travel-specific benchmarks such as TripTailor (Shen et al., 2025), ChinaTravel (Shao et al., 2025), and Flex-TravelPlanner (Oh et al., 2025) move toward user-adaptive planning but remain largely single-turn or omit explicit modeling of constrained optimization. Concurrent work (Qian et al., 2025) also explores user preference optimization, though with a primary focus on elicitation rather than adaptive constraint satisfaction. Our benchmark uniquely combines tool calling, planning, and preference optimization in a multi-turn setting—capturing the progressive revelation of user constraints and the agent's need to dynamically adapt its reasoning and optimization strategies.

Multi-turn interaction research further underscores the importance of modeling incremental user intent and adaptive reasoning for real-world deployment (Laban et al., 2025; Abdulhai et al., 2023; Wan et al., 2025). Complementary work in task-oriented dialogue (Budzianowski et al., 2018; Rastogi et al., 2019) provides foundational insights into how users iteratively refine goals and reveal constraints over multiple turns, directly informing the design of our user simulator. In App. B, we include an extended review of related work.

## 3 *COMPASS* BENCHMARK

### 3.1 TRAVEL PLANNING AS CONSTRAINED PREFERENCE OPTIMIZATION

A central challenge for real-world agents is not only to satisfy user requirements but also to optimize for user satisfaction. To capture this duality, we formulate travel planning as a constrained preference optimization problem. Formally, a general constrained optimization problem can be expressed as:

$$\min_{x} f(x) \tag{1}$$

$$\text{s.t.} \quad g_i(x) = c_i, \ i = 1, \ldots, n \tag{2}$$

$$h_j(x) \geq d_j, \ j = 1, \ldots, m \tag{3}$$

In travel scenarios, **hard constraints** represent non-negotiable requirements such as budget limits ("total cost under \$3000"), occupancy needs ("6 people, max 2 per room"), or logistical rules ("permit required for Extended Cave Tour"). These constraints define the feasible set of solutions—any proposed itinerary $x'$ must satisfy Eqs. (2)–(3). Within this feasible region, **soft preferences** capture user priorities like minimizing total cost or maximizing amenities. The optimization objective $f(x)$ reflects user utility, ranking feasible solutions by their desirability. This separation between *feasibility* and *optimality* provides a principled framework for evaluating whether agents can go beyond finding any feasible plan ($x'$) to achieving near-optimal satisfaction ($f(x') \approx f(x^*)$).

### 3.2 TASK DESIGN.

**Optimization Types.** To balance realism with clarity of task specification, we design two complementary preference optimization objectives $f(x)$ that capture distinct user behaviors.

- **Single Metric Optimization.** The user specifies one numerical objective: "find the cheapest trip" (minimize cost) or "book the highest-rated hotel" (maximize review score). Task success requires optimizing $f(x)$ within the feasible set defined by $g_i(x)$ and $h_j(x)$.
- **Feature Count Maximization.** The user provides a wish list of desirable attributes: "hotel with spa, pool, and kitchen; direct flight with WiFi." These are flexible preferences and satisfying three features is better than one. The agent maximizes $f(x)$, where $f(x)$ counts satisfied optional attributes while still respecting the same feasibility constraints. These tasks reflect realistic "nice-to-have" preferences.

Table 1: **COMPASS benchmark characteristics**. *Left:* Task distribution (Sec. 3.2) across three complexity levels, and split between the two optimization types, and data scale covering geographic coverage, inventory size, and benchmark components. *Right:* Complete tool suite (Sec. 3.3) organized by category, listing 18 APIs for navigation, search, validation, and utility functions.

| Category | Item | Count |
|---|---|---|
| *Task Levels* | Level I (Hotel only) | 142 |
| | Level II (Hotel, Flight) | 69 |
| | Level III (Hotel, Flight, Permit) | 70 |
| | **Total Tasks** | **281** |
| *Task Types* | Single Metric Optimization | 154 |
| | Feature Count Maximization | 127 |
| *Database Scale* | National Parks (Destinations) | 20 |
| | Airports | 21 |
| | Hotel Room Bookable Offers | 100,000+ |
| | Flight Bookable Offers | 67,000+ |
| | Trail Permits | 50 |

| Tool Category | Tool Name |
|---|---|
| *Hotel* | search_hotel |
| | get_hotel_details |
| | get_room_details |
| | search_location_name |
| *Flight* | search_airports |
| | search_flights |
| | get_flight_details |
| | get_airport_location |
| *Permit* | list_permits |
| | search_permit_availability |
| *Utility* | notebook |
| | calendar |
| | calculator |
| | calculate_driving_distance |
| | validate_hotel_(_flight_permit)_id |

These objectives are quantitatively measurable, ensuring that optimization objectives are always well-specified. Combined with hard constraints, they yield realistic behaviors: a user may seek the cheapest option (*optimization objective*) while requiring three-star hotels (*hard constraint*), or prefer business-class and/or direct flights (*optimization objective*) while limiting spending to $2,000 (*hard constraint*).

**Difficulty Levels.** For each task optimization type, we design three difficulty levels.

- **Level I (Hotel Only):** Agents must navigate room configurations and pricing structures under constraints such as budget, occupancy, and amenities.
- **Level II (Hotel + Flight):** Agents plan complete trips with both accommodations and transportation. This requires temporal reasoning to align departure dates with hotel availability while distributing the budget across both services.
- **Level III (Hotel + Flight + Permits):** Agents incorporate reservations for guided tours or hiking experiences with limited availability. Permits introduce additional temporal constraints: permit availability can dictate travel dates, which in turn determine feasible hotel and flight options.

This progression serves three evaluation purposes: (1) *Sequential reasoning*: higher levels require chaining constraints across domains, such as permit availability impacting hotel selection, which then constrains flight booking. (2) *Information gathering complexity*: agents must use more diverse tools to collect information from multiple services (e.g., hotel APIs, flight APIs, permit systems) and synthesize this information to produce coherent plans. (3) *Search space expansion*: the combinatorial solution space grows substantially with each level, as agents must consider all valid combinations of options across multiple services (e.g., Level III requires exploring hotel × flight × permit combinations).

**Scenario Diversity.** We design tasks using 36 diverse user templates, each specifying a travel scenario (e.g., backpacker, family with children, wedding) with associated hard constraints and optimization objectives. From these templates, we generate 240 total tasks across the three difficulty levels by sampling parameters such as travel dates, destinations, and budget ranges (See Tab. 1 for task breakdown). App. C.1 provides a detailed breakdown of all tasks and concrete examples.

## 3.3 DATABASE CONSTRUCTION & TOOLS DESIGN

The complexity of the underlying data directly shapes the difficulty and realism of an optimization benchmark. Oversimplified databases fail to reflect real-world challenges, while overly complex ones reduce practical relevance. To balance these factors, *COMPASS* uses real travel data obtained through RapidAPI,[1] constructing clean SQL databases from actual booking information. Our dataset covers 20 U.S. National Parks, each with 20 hotels offering diverse room types, prices, amenities, and policies. We also include flight data from four major airlines with varied routing options and booking classes. Permit data is mocked, but modeled after real high-demand permits (e.g., the Yosemite Half Dome hike , ensuring scenarios remain authentic. In App. F.1, we describe how we built the database,

---

[1] https://rapidapi.com/DataCrawler/api/booking-com15

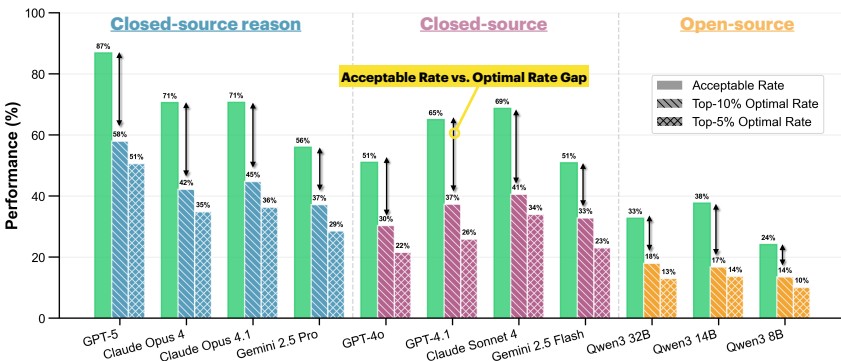

Figure 2: *COMPASS* **benchmark main results. Acceptable rate** measures feasibility (satisfying all hard constraints). **Optimal rate** measures preference optimization (achieving utility within the top 10% of feasible solutions). All models show a $\sim 20\%$ gap between high acceptable rates and low optimal rates, revealing that agents settle for feasible solutions rather than optimizing preferences. Encouragingly, open-source models like Qwen3-32B achieve non-trivial performance, demonstrating emerging agentic capabilities.

and in App. F.2, we summarize key statistics for the hotel and flight data. We also highlight examples from real-world datasets that capture nuances and complexities often missing in synthetic data.

*COMPASS* also provides a comprehensive tool suite that mirrors commercial booking platforms. Core search functions (search_hotel, search_flights) are complemented by utility tools (calculator, notebook) that support multi-step problem solving. Our search tools accept rich filters (e.g., price ranges, amenity requirements, routing preferences), providing agents the optionality to strategically conduct search. A full list of tools is shown in Tab. 1 *(right)* and detailed in App. D.

### 3.4 USER SIMULATOR DESIGN

We use GPT-5 as the backbone of our user simulator. Unlike prior benchmarks (Yao et al., 2024; Lu et al., 2024) that rely on a single static prompt for the entire dialogue, our simulator employs **dynamic prompting**—adjusting instructions both across different users and within each conversation turn. Compared to existing works (Yao et al., 2024; Lu et al., 2024) that use static user prompt designs, our simulator enables fine-grained control over user behavior. We control four major-axis: *within-conversation dynamics*: **(1) Progressive constraint revelation**, where users disclose task information incrementally rather than upfront, and **(2) Trust and communication patterns**, where users vary in whether they accept recommendations immediately or demand explicit reasoning. We also introduce two axes of *persona diversity*: **(3) Constraint-checking reliability**, where users differ in how consistently they detect and correct agent mistakes, and **(4) Conversation style**, where personas range from concise and pragmatic to repetitive, emotional, or insistent. The 4 axis controls orthogonal aspects of user behaviors and in total we have 108 distinct user types. In App. E, we provide extensive details on each of the 4 axis as well as the dynamic prompting strategy.

### 3.5 EVALUATION METRIC DESIGN

**Ground Truth Generation.** To evaluate agents reliably, we construct ground-truth solutions for every task via exhaustive search. We enumerate all candidate hotels, flights (Levels II & III), and permits (Level III), apply domain-specific filters (e.g., direct flights, 4-star hotels), and generate all cross-domain combinations. Combinations violating any hard constraints Eq. (2)-(3) are discarded, and the remaining set defines the feasible space. For each task, we compute an exact utility score $f(x)$. In single-metric tasks, $f(x)$ corresponds to the target metric (e.g., total price, average rating); in feature-count tasks, $f(x)$ is the number of satisfied desired features. This exhaustive enumeration guarantees correctness, since every feasible solution is scored. We further filter tasks with too few feasible solutions or excessive ties ($\geq 20\%$ of feasible set) to ensure meaningful optimization difficulty.

**Evaluation Metrics.** We evaluate each agent along two complementary dimensions:

Table 2: **Human evaluation of LLM user simulator quality.** *Left:* Scoring rubrics used by annotators for *clarity* and *contextual appropriateness*. *Right:* Human evaluation results on 198 user responses. Rubric scores are reported as mean $\pm$ standard deviation, as well as median, and quantile cut-offs. Error rates are reported as percentages of responses exhibiting each error type.

| Clarity |
| --- |
| **5:** Crystal clear communication of requirements and goals |
| **4:** Clear with no confusion about the task |
| **3:** Mostly clear but some minor ambiguity that won't affect results |
| **2:** Unclear communication that could lead agent to make mistakes |
| **1:** So unclear the agent will likely fail the task |
| **Contextual Appropriateness** |
| **5:** As good as you would write or better |
| **4:** Good without any obvious issues |
| **3:** A bit unnatural but overall harmless to the conversation flow |
| **2:** Noticeable issues that do not follow the conversation logic |
| **1:** Response is completely disconnected from the logic |

| Metric | Result |
| --- | --- |
| **Quality Scores (1–5)** ↑ | |
| **Clarity** | 3.9 ± 1.2 |
| Median | 4.0 |
| Percentiles (25/50/75/90/95) | 3 / 4 / 5 / 5 / 5 |
| **Contextual appropriateness** | 3.7 ± 1.1 |
| Median | 4.0 |
| Percentiles (25/50/75/90/95) | 3 / 4 / 5 / 5 / 5 |
| **Error Rates** | |
| Factual hallucination | 6.1% |
| Information revelation failures | 4.0% |
| Constraint checking failures | 0.5% |

**Acceptable Rate:** Fraction of tasks where the agent's recommendation $x'$ satisfies all hard constraints, i.e., $x' \in \{x : g_i(x) = c_i, h_j(x) \geq d_j\}$.

**Optimality Rate:** Fraction of tasks where $x'$ satisfies all constraints and ranks within the top-$i^{\text{th}}$ percentile of feasible solutions under $f(x)$, measuring how close the agent's recommendation is to the global optimum $f(x^*)$.

## 4 EXPERIMENT

### 4.1 MAIN RESULTS

We evaluate *COMPASS* on a range of frontier closed-source models, including both reasoning-enabled and standard versions of GPT (OpenAI et al., 2023; OpenAI, 2025), Claude (Anthropic, 2025), and Gemini (Comanici et al., 2025), as well as the frontier open-source model Qwen3 Yang et al. (2025). All models support native tool calling, and we pass the tool schema directly to the model interface. We evaluate each agent on the final itinerary it produces. Fig. 2 reports aggregate performance. App. G provides inference details for each model, App. A.1 reports a detailed breakdown table of model performances, and App. A.2 confirms benchmark stability with small error bars across repeated runs.

**Acceptable-Optimal Gap.** We find a significant **gap between constrained-preference satisfaction and preference optimization**. All models achieve high acceptable rates, showing they can interpret constraints and return feasible solutions. However, their 5% and 10% optimality rates are substantially lower, revealing that agents often settle for feasible but suboptimal solutions even when objectives are simple and explicitly stated. Reasoning models outperform non-reasoning counterparts, and open-source Qwen3-32B (reasoning enabled) reaches close to 20% optimality rate—comparable to some closed-source systems such as GPT-4o. This suggests that strong agentic capabilities are emerging in open-source models. Nonetheless, GPT-5 shows superior performance across both dimensions, outperforming all other models.

**Conversation Efficiency.** Beyond final performance, efficiency is critical for user-facing agents. We measure it using our user simulator (Sec. 3.4): for each conversation, we record when the user has revealed all information about a task ($t^*$) and count extra turns ($\Delta t$) needed to deliver the final recommendation. Extra turns arise when agents fail to provide recommendations or propose constraint-violating options. Fig. 3 **(D)** shows GPT-5 excels in both success and conversation efficiency, requiring the fewest post-revelation turns. Notably, efficiency differs even among models with similar success rates: Gemini-2.5-Flash takes far more turns than GPT-4o despite comparable outcomes. In App. A.7, we confirm that the observed difference is not result of simulator variance.

### 4.2 VALIDATING LLM USER SIMULATOR QUALITY

To ensure our LLM user simulator produces realistic and reliable interactions, we randomly sample 45 full conversations spanning diverse personas, task types, and agent interactions. From these, we extract **198** individual user responses for human evaluation by an independent expert annotation service, covering the full range of scenarios in the *COMPASS* benchmark.

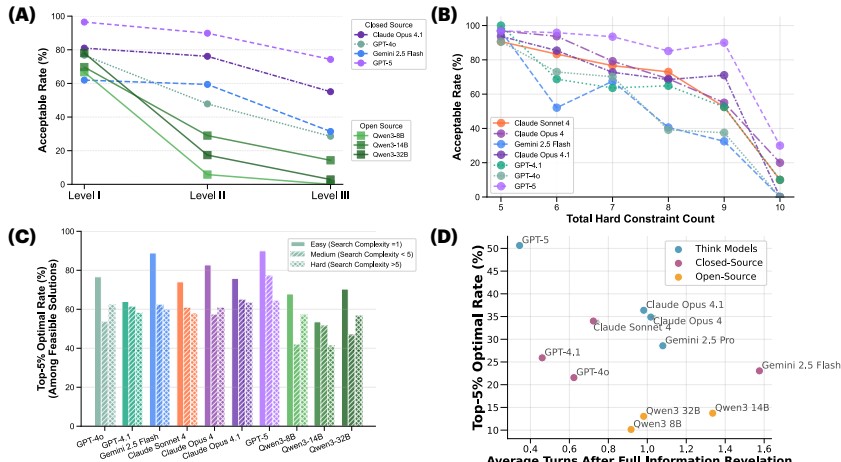

Figure 3: **Performance breakdown across benchmark dimensions. (A)** Performance degrades with increasing plan coordination complexity (Levels II–III), with open-source models showing especially steep declines (*green*). **(B)** Constraint satisfaction rates drop as the number of hard constraints increases, with only the strongest models handling 8+ constraints reliably. **(C)** Preference optimization weakens as search complexity grows (more searches required to reach the ground-truth optimum). **(D)** Conversation efficiency analysis: how fast agent achieves solutions with the fewest post-information revelation turns.

Annotators assess each response along two dimensions. **(1) Rubric-based quality assessment:** responses are scored on two axes. The *clarity* axis measures whether user messages are expressed in a way that avoids misleading the agent or causing task failure; a score below 3 (on a 1–5 scale) indicates that the simulator may have provided confusing or incorrect task information. The *contextual appropriateness* axis evaluates whether responses flow naturally within a travel-planning dialogue. **(2) Detailed error detection:** since behaviors such as constraint revelation and trust patterns are explicitly controlled (Sec. 3.4), annotators tag whether the simulator follows its prompts correctly without hallucination or deviation. The scoring rubric appears in Tab. 2 *(left)*, with full annotation instructions and error definitions in App. H.

Results (Tab. 2, *right*) show high quality scores (median clarity: 4; contextual appropriateness: 4 on a 1–5 scale) and very low error rates—4% for constraint revelation and 0.5% for feedback accuracy. These results confirm that the simulator reliably follows its script while producing clear, natural responses, validating it as a realistic tool for benchmarking agent capabilities. In App. H.2, we show human evaluation breakdown across different user types. Overall, we do not observe significant differences across user types.

# 5 TASK PERFORMANCE ANALYSIS

## 5.1 WHY DO AGENTS STRUGGLE IN CONSTRAINED OPTIMIZATION?

We now investigate why agents either fail to satisfy constraints or produce suboptimal solutions.

**Task Difficulty: Multi-Component Coordination.** As task level increases, agents must gather information from diverse tools, reason about cross-component dependencies (e.g., flight arrivals constraining hotel check-in), and jointly optimize while satisfying constraints spanning multiple services. Fig. 3 **(A)** shows performance degrades sharply as coordination demands increase. While most models handle single-component tasks reliably (Level I), performance drops substantially on Level II and collapses on Level III. Open-source models perform competitively with frontier systems on Level I but degrade far more steeply on Levels II–III, revealing that cross-component constraint propagation and coordinated optimization remain critical limitations, particularly for open-source models.

**Constraint Complexity: Managing Multiple Requirements.** We now examine how agents handle increasing numbers of hard constraints. Each task includes at least five fundamental constraints (loca-

Table 3: **Tool call error rates.** Fraction of calls (%) containing tool name, parameter, or content hallucinations. Closed-source models are nearly error-free, whereas open-source models frequently hallucinate parameter names.

| Model | Name | Param | Content | Total |
|-------|------|-------|---------|-------|
| Claude Sonnet 4 | 0.03 | 0.00 | 0.14 | 0.17 |
| GPT-4o | 0.00 | 0.00 | 0.21 | 0.21 |
| Claude Opus 4 | 0.16 | 0.00 | 0.25 | 0.41 |
| GPT-4.1 | 0.00 | 0.32 | 0.42 | 0.74 |
| GPT-5 | 0.00 | 0.04 | 1.41 | 1.45 |
| Gemini 2.5 Pro | 0.16 | 0.04 | 1.38 | 1.58 |
| Gemini 2.5 Flash | 0.00 | 2.36 | 0.10 | 2.46 |
| Qwen3-14B | 0.00 | 4.88 | 0.72 | 5.60 |
| Qwen3-32B | 0.00 | 5.00 | 3.85 | 8.85 |
| Qwen3-8B | 0.14 | 10.81 | 1.08 | 12.03 |

tion, dates, room occupancy, budget, number of travelers), with additional requirements (amenities, policies, accessibility) further restricting the feasible set. Fig. 3 (**B**) shows constraint satisfaction degrades as requirement count increases. GPT-5 and Claude maintain high success with 8+ constraints, while most models drop sharply beyond 7. Critically, this persists even after users explicitly state requirements and correct violations, indicating that constraint satisfaction, which is a prerequisite for feasibility, remains a challenge to weaker models.

**Optimization Complexity: Search Space Difficulty.** Beyond satisfying constraints, agents should identify the best option among feasible solutions. We measure optimization difficulty through *search complexity*: the minimum number of distinct tool calls required to guarantee discovering the ground-truth optimum. For example, "cheapest weekend in August at Yosemite" requires searching all four weekends (complexity = 4), while "August 15-17 at Yosemite, cheapest hotel" requires one search (complexity = 1). Fig. 3 (**C**) analyzes optimality rates conditional on producing feasible solutions—measuring how often valid recommendations fall within top 5% optimality. When search complexity is minimal (1 search), agents achieve high optimality. As complexity increases to medium ($<5$) and hard ($>5$ searches), optimal rates decline across all models. Even GPT-5 drops from 80% on easy tasks to 60% on hard tasks, demonstrating that agents often fail to explore systematically enough to identify truly optimal choices—explaining the acceptable–optimal gap (Sec. 4.1).

# 6    SMART TOOL USE FOR CONSTRAINED OPTIMIZATION

## 6.1    TOOL USE AND INFORMATION GATHERING ANALYSIS

**Tool Call Correctness.** We categorize tool call mistakes into three types: **tool name hallucination** (calling a non-existent tool), **parameter hallucination** (using an invalid argument name), and **content hallucination** (fabricating parameter values not present in the database). In Tab. 3, we report error rates for all three categories. Closed-source models almost never hallucinate tools or parameters ($< 1\%$ total), showing strong tool call capabilities. In contrast, we see high error rates in open-source models (up to 12%), driven mainly by parameter hallucinations. This results indicate that open-source models still lack fundamental tool-calling reliabilities.

**Search Extensiveness.** Fig. 4 examines how tool call frequency correlates with task success. For each model, we bin tasks by the number of tool calls used and report top-5% optimality rates within each bin. Better-performing models make more tool calls: GPT-5 frequently calls 21+ tools while weaker models typically make fewer than 10. Examining performance within bins, we see that GPT-5 maintains high optimality even with 21+ calls, demonstrating robust multi-step reasoning and information processing capability. GPT-4o degrades as call frequency increases, while open-source models like Qwen3-8B rarely attempt extended search, indicating limited capacity for the extensive exploration stronger models perform routinely. Tool call frequency measures search effort but not search quality. To understand what models actually discover, we next analyze the information content retrieved through these calls.

**Information Gathering Quality.** While tool call frequency indicates search effort, it does not guarantee information diversity. For example, models may issue many calls that retrieve redundant results or use poorly-targeted parameters. We therefore quantify actual information content by

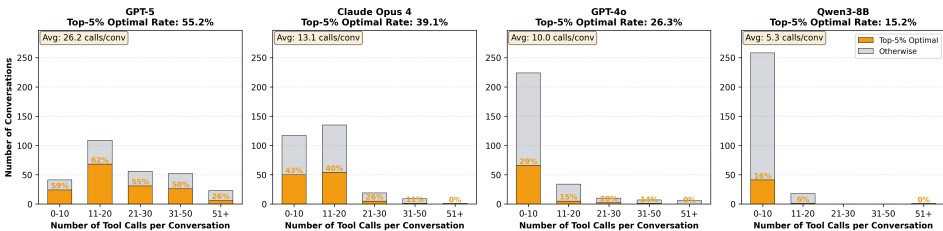

Figure 4: **Tool call frequency and optimization success.** For each bin of tool call counts, we show the distribution of tasks and corresponding top-5% optimality rate. Stronger models make more calls on average and maintain high performance under extended search. Weaker models either fail to explore deeply (open-source) or show performance degradation when attempting complex multi-step reasoning (GPT-4o).

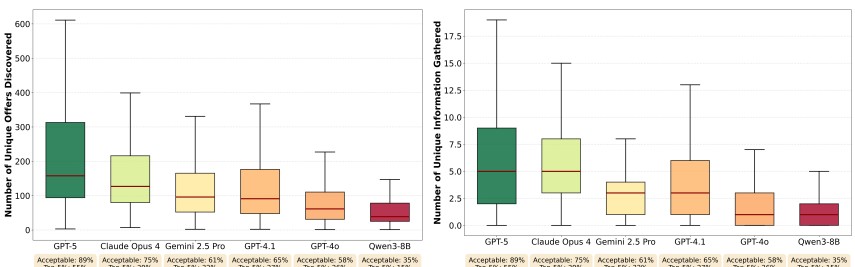

Figure 5: **Information gathering and task success.** *Left:* Number of unique bookable offers discovered (each identified by a distinct offer ID). *Right:* Number of unique auxiliary information items retrieved (amenities, policies, distances, etc.). Success strongly correlates with gathering more information, indicating current models remain in a regime where broader exploration universally improves outcomes.

categorizing tool-acquired data into two types: (1) *Bookable offers*: these are concrete options (hotel rooms, flights, permits) uniquely identified by offer IDs encoding specific attributes, analogous to confirmation numbers. We replay tool calls and count unique offer IDs to measure solution space coverage. (2) *Auxiliary information*: these are contextual details retrievable via additional tools (amenities, pet policies, transfers, distances), counted similarly. Fig. 5 shows that models discovering more unique offers and auxiliary details are substantially more likely to produce top-5% optimal solutions. This demonstrates that exploring a larger search space enables better optimization.

However, this correlation reveals current models remain in a regime where *more information universally improves performance*. Ideally, agents should gather *sufficient but not excessive* information to identify optimal solutions without over-exploration. While gathering too little hurts performance (as weaker models demonstrate), gathering excessively wastes test-time compute and increases latency. GPT-5 currently dominates through comprehensive exploration, but future models should prioritize precision and efficiency: achieving high success while minimizing unnecessary information retrieval.

## 6.2 CASE STUDY

While our ground truth relies on exhaustive search (Sec. 3.5), real travel planning requires reasoning, strategy, and efficient tool use. To examine whether agents display such behaviors, we analyze their tool usage and reasoning on a representative Level II task, asking them to explain their thought process (Fig. 6, *top*).

**Strategic vs. Naive Planning.** GPT-5 exhibits strategic behavior (Fig. 6, *bottom left*). Rather than searching blindly, it checks the calendar, targets weekday departures (anticipating midweek price drops), and samples across August to ensure coverage. This heuristic aligns with our data—midweek trips are indeed cheaper on average—but whether it is "smart" depends on context, as models may apply incorrect heuristics elsewhere. A more capable agent would either (i) *verify* such assumptions or (ii) *articulate* them to the user for confirmation. GPT-5 does the latter, making its reasoning trans-

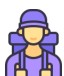

Hi, I'm looking to plan a trip from JFK to Yellowstone National Park for one person. I want the absolute *cheapest* option (hotel and flight combination) for a 5-day trip, including flying time, which will cover 4 nights in a hotel. I'm completely flexible on the dates as long as it's sometime in August 2025. My total budget for flights and hotel combined is $2,200 or less. Can you help me arrange this? I also want you to explain your reasoning behind how you solve this problem. Please explain your search process and justify why you will the tools in what order and how to pass in the arguments.

**GPT-5 Tool Call Trace**

[TOOL CALL #1] **search_airports** {'park_name': 'Yellowstone National Park'}
[TOOL CALL #2] **get_weekday** {'date_str': '2025-08-24'}
[TOOL CALL #3] **get_weekday** {'date_str': '2025-08-28'}
[TOOL CALL #4] **search_flights** {'origin': 'JFK', 'destination': 'BZN', 'departure_date': '2025-08-24', 'return_date': '2025-08-28', 'passengers': 1, **'max_price': 2200**}
[TOOL CALL #5] **search_flights** {'origin': 'JFK', 'destination': 'JAC', 'departure_date': '2025-08-24', 'return_date': '2025-08-28', 'passengers': 1, 'max_price': 2200}
*[skipping additional **search_flights** calls...]*
[TOOL CALL #11] **search_hotel** {'location': 'West Yellowstone', 'start_date': '2025-08-24', 'end_date': '2025-08-28', 'num_guests': 1, **'max_price_nightly': 300**, 'min_review_score': 7, **'free_cancellation': True**}
*[skipping additional alternating **search_flights** and **search_hotel** calls...]*

*Justification:* - They meet your 5-day/4-night constraint cleanly: **Tue-Sat** blocks give exactly four hotel nights (Tue-Fri) and avoid paying for a typically pricier Saturday night.
- **Mid-week pricing advantage**: Flying and staying mid-week (Tue-Thu/Fridays) is usually cheaper than Friday night/Saturday/Sunday, especially in gateway towns around Yellowstone during peak season.
- Broad coverage across August: Sampling **early, mid, and late August** catches typical fare/hotel volatility without brute-forcing every single date. This helps surface the reliably cheapest week while keeping the search efficient.

**Claude Sonnet 4 Tool Call Trace**

[TOOL CALL #1] **search_airports** {'park_name': 'Yellowstone National Park'}
[TOOL CALL #2] **search_flights** {'origin': 'JFK', 'destination': 'COD', 'departure_date': '2025-08-05', 'return_date': '2025-08-10', 'passengers': 1, **'max_price': 1500**}
[TOOL CALL #3] **search_flights** {'origin': 'JFK', 'destination': 'COD', 'departure_date': '2025-08-12', 'return_date': '2025-08-17', 'passengers': 1, 'max_price': 1500}
[TOOL CALL #4] **search_location_name** {'query': 'Yellowstone National Park'}
[TOOL CALL #5] **search_hotel** {'location': 'Yellowstone National Park', 'start_date': '2025-08-05', 'end_date': **'2025-08-09'**, 'num_guests': 1, **'max_price_nightly': 400**}
[TOOL CALL #6] **calculate** {'expression': '308 + 888.21'}

*Justification*: This itinerary gives you exactly what you requested: 5 days total including travel time, with 4 nights of accommodation near Yellowstone. The Yellowstone Regional Airport is only 1 hour drive from the park, and your hotel is very close to the park entrance. You'll have plenty of budget left over for meals, park entrance fees, and activities!

Figure 6: **Case study of tool calls and reasoning traces.** *Top:* Prompt given to models for a Level II task, with explicit reasoning requested. *Bottom:* GPT-5 (*left*) demonstrates strategic planning by avoiding weekends, systematically exploring date ranges, and using optional parameters (e.g., price filters) to narrow searches. Claude-Sonnet-4 (*right*) applies optional parameters but searches only two arbitrary dates without justification. It also makes a temporal coordination error by misaligning hotel and flight dates.

parent and allowing user-guided refinement. Nevertheless, it also adds unjustified parameters—e.g., enforcing `free_cancellation=True`—which may exclude valid cheaper options, illustrating the trade-off between strategic search and fidelity to user intent. Claude Sonnet-4 (Fig. 6, *bottom right*) behaves more naively: it searches only two dates without justification and commits a major coordination error by misaligning hotel and flight dates. Its reasoning traces are generic ("*gives exactly what you requested*"), concealing such mistakes.

These results show that intelligent travel planning requires more than constraint satisfaction or exhaustive search; it depends on heuristics and reasoning grounded in tool use. We expect future agents to employ richer strategies, such as maintaining a running "best" option and progressively filtering around it, balancing efficiency with optimality. Overall, our case studies demonstrate that COMPASS evaluates reasoning-driven planning behaviors beyond mere tool execution.

# 7 CONCLUSION AND DISCUSSION

We introduced COMPASS, a benchmark for constrained-preference optimization in realistic travel planning. Our evaluation shows that while frontier agents succeed at satisfying constraints, they struggle with multi-step planning and preference optimization in multi-turn settings, highlighting a fundamental gap in current capabilities.

**Limitations & Future Directions.** OMPASS currently focuses on travel planning and a simulated user; extending to broader tasks (e.g., cars, restaurants) and to web-based agents interacting with real interfaces would increase ecological validity. Our simulator, though controllable and reliable, cannot yet capture the full unpredictability of human dialogue. Future work should expand to other domains (e-commerce, healthcare, finance), incorporate richer user modeling (human-in-the-loop or multi-agent), and evaluate agentic workflows that go beyond native tool calling to test whether structured orchestration enhances capabilities. The planning complexity we observe suggests agents remain far from robust real-world deployment; solving these tasks will require not only constraint satisfaction but also intelligence and reasoning in natural, non-mathematical forms.

## REPRODUCIBILITY STATEMENT

Upon publication, we will release the full *COMPASS* benchmark, including code, task definitions, and data. Detailed descriptions of task construction and dataset statistics are provided in Appendix C.1, while experiment settings and evaluation procedures are documented in Appendix G.

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

CONTENTS

# A ADDITIONAL RESULTS

## A.1 FULL BENCHMARK RESULT

In Table 4, we report the benchmark performance in table form, including a detailed performance breakdown by three task levels.

Table 4: **Model performance breakdown by complexity level on *COMPASS* benchmark**. We evaluate our benchmark across three complexity levels: **Level I** (hotel-only tasks, ∼142 tasks), **Level II** (flight+hotel tasks, ∼69 tasks), and **Level III** (complex tasks with permits, ∼70 tasks). Performance generally degrades with increased complexity, with open-source models showing dramatic capability gaps in multi-step planning.

| | Model | Level | Optimal Rate (%) ↑ | | | Acceptable Rate (%) ↑ |
|---|---|---|---|---|---|---|
| | | | Top-5 | Top-10 | Top-20 | |
| **Think** | GPT-5 | I | 68.3 | 77.5 | 88.0 | 96.5 |
| | | II | 52.2 | 56.5 | 69.6 | 89.9 |
| | | III | 31.4 | 40.0 | 48.6 | 74.3 |
| | | Total | 50.6 | 58.0 | 68.7 | 86.9 |
| | Claude Opus 4 | I | 51.4 | 63.4 | 77.5 | 89.4 |
| | | II | 26.1 | 27.5 | 39.1 | 68.1 |
| | | III | 27.1 | 35.7 | 38.6 | 54.3 |
| | | Total | 34.9 | 42.2 | 51.7 | 70.6 |
| | Gemini 2.5 Pro | I | 42.3 | 52.1 | 62.7 | 76.8 |
| | | II | 20.6 | 29.4 | 30.9 | 41.2 |
| | | III | 22.9 | 30.0 | 34.3 | 50.0 |
| | | Total | 28.6 | 37.2 | 42.6 | 56.0 |
| **Closed** | GPT-4o | I | 40.1 | 53.5 | 69.0 | 76.8 |
| | | II | 17.4 | 23.2 | 30.4 | 47.8 |
| | | III | 7.1 | 14.3 | 14.3 | 28.6 |
| | | Total | 21.6 | 30.3 | 37.9 | 51.1 |
| | GPT-4.1 | I | 31.7 | 44.4 | 56.3 | 66.9 |
| | | II | 23.2 | 33.3 | 42.0 | 71.0 |
| | | III | 22.9 | 34.3 | 41.4 | 57.1 |
| | | Total | 25.9 | 37.3 | 46.6 | 65.0 |
| | Claude Sonnet 4 | I | 43.0 | 54.2 | 69.7 | 83.8 |
| | | II | 34.8 | 36.2 | 49.3 | 66.7 |
| | | III | 24.3 | 31.4 | 40.0 | 55.7 |
| | | Total | 34.0 | 40.6 | 53.0 | 68.7 |
| | Gemini 2.5 Flash | I | 31.7 | 42.3 | 52.8 | 62.0 |
| | | II | 21.7 | 36.2 | 40.6 | 59.4 |
| | | III | 15.7 | 20.0 | 24.3 | 31.4 |
| | | Total | 23.0 | 32.8 | 39.2 | 50.9 |
| **Open** | Qwen3 8B | I | 30.4 | 39.1 | 49.3 | 66.7 |
| | | II | 0.0 | 1.4 | 1.4 | 5.8 |
| | | III | 0.0 | 0.0 | 0.0 | 0.0 |
| | | Total | 10.1 | 13.5 | 16.9 | 24.2 |
| | Qwen3 14B | I | 26.8 | 34.5 | 47.9 | 69.7 |
| | | II | 8.7 | 8.7 | 11.6 | 29.0 |
| | | III | 5.7 | 7.1 | 10.0 | 14.3 |
| | | Total | 13.7 | 16.8 | 23.2 | 37.7 |
| | Qwen3 32B | I | 33.3 | 46.8 | 61.0 | 78.0 |
| | | II | 4.3 | 4.3 | 7.2 | 17.4 |
| | | III | 1.4 | 2.9 | 2.9 | 2.9 |
| | | Total | 13.0 | 18.0 | 23.7 | 32.8 |

## A.2 BENCHMARK STABILITY

To demonstrate the stability and reliability of our *COMPASS* benchmark, we evaluate Claude-4-Sonnet across 5 independent runs on the full benchmark suite (Table 5). Each run uses identical experimental settings but different random seeds for task sampling and user simulator behavior. The consistency across runs demonstrates that our user simulator produces reproducible interactions despite its stochastic nature, and that the evaluation metrics accurately capture systematic differences in agent performance rather than measurement noise.

## A.3 SENSITIVITY TO DIVERSE USER TYPES

LLM agents interact with users who vary widely in how they communicate task requirements, express trust, and articulate preferences. In App. E, we detail the major design axis that impacts how user interact with the agent. Here, to understand how these behavioral differences influence agent

Table 5: **Benchmark stability.** Benchmark stability across 5 independent runs with Claude-4-Sonnet. Results show low variance, demonstrating reliable measurement of agent capabilities.

| Metric | Run 1 | Run 2 | Run 3 | Run 4 | Run 5 | Mean (Std) |
|---|---|---|---|---|---|---|
| Optimal Top-5 | 33.97 | 34.96 | 33.44 | 31.92 | 34.43 | 33.74 (1.04) |
| Optimal Top-10 | 39.76 | 41.60 | 40.56 | 39.44 | 41.04 | 40.48 (0.80) |
| Optimal Top-20 | 47.44 | 53.36 | 54.96 | 52.80 | 54.40 | 52.59 (2.69) |
| Solve Rate | 68.80 | 68.80 | 67.20 | 68.80 | 66.08 | 67.94 (1.12) |

performance, we systematically vary these of user behavior axis in our simulator and analyze model robustness along each axis and report how these user-side factors affect task outcomes.

**Information Revelation Speed.** One key axis in our user simulator controls the rate at which users disclose task information (i.e., the full set of hard constraints). This dimension captures the contrast between users who immediately provide all requirements and those who incrementally introduce new constraints over the course of the conversation. To assess whether delayed information disclosure impairs model performance, we group conversations by the number of turns required for the user to fully reveal the task specification and measure the resulting Top-5Across both closed-source and open-source models, we observe a clear degradation in performance as the number of revelation turns increases. Prior work (Laban et al., 2025) reported similar effects in more controlled domains such as math and synthetic reasoning benchmarks; our results extend this phenomenon to significantly more realistic, multi-constraint tasks (e.g., booking itineraries, planning logistics), highlighting that slow or delayed information revelation remains a persistent challenge for current LLMs.

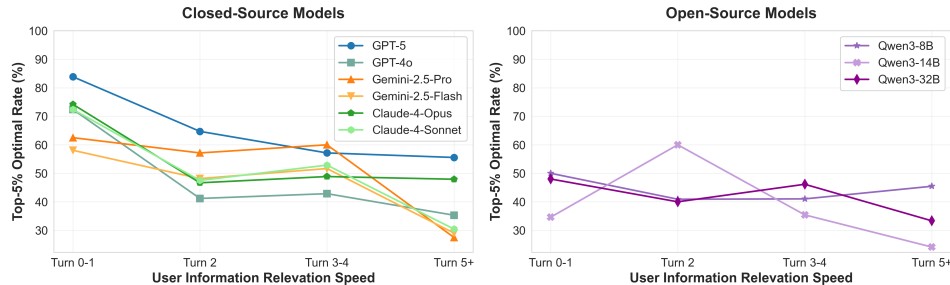

Figure 7: **Effect of information revelation speed on model performance.** Top-5% Optimal Rate as a function of how many dialogue turns the user requires to fully reveal all task constraints. Higher revelation latency is associated with consistently lower performance across models, indicating that incremental or delayed specification of requirements poses a substantial challenge for current models.

**Trusting Levels.** Another key user simulator design axes controls the user's trust level—distinguishing between *trusting* users, who readily accept the agent's recommendations, and *suspicious* users, who prompt the agent to double-check its own output. We analyze whether this questioning behavior elicits any form of self-reflection that improves task performance. To compare model outcomes across the two user types, we report the top-5% optimal rate for each group and perform a two-proportion $z$-test to assess statistical significance ($p \leq 0.05$). As shown in Tab. 6, prompting the agent to re-evaluate its answer—without providing additional feedback—does *not* significantly improve performance for any model tested.

**Consistency Across Communication Styles.** An ideal model should maintain stable performance regardless of user communication style (e.g., rude, polite, formal, or casual). Our user simulator includes nine distinct communication styles (App. E, Tab. 10). To assess robustness, we measure each model's Top-5% Optimal Rate across these groups and quantify consistency using the Coefficient of Variation (CV)—the standard deviation of performance across styles, normalized by the mean. Lower CV values indicate more consistent performance. Fig. 8 illustrates examples of per-style variation and corresponding CV computation, and Tab. 7 reports the CV for all models. Results show that GPT-5 achieves the highest consistency across user styles, while both open-source and several frontier models exhibit greater sensitivity to stylistic variation.

Table 6: **Effect of user trust level on model performance.** Top-5% optimal rates are compared between suspicious (questioning) and trusting users. The $p$-values are computed via a two-proportion $z$-test. Across models, the differences are not statistically significant, suggesting that simply asking the model to "double-check" its own reasoning does not reliably enhance task outcomes.

| Model | Suspicious | Trusting | $p$-value |
|---|---|---|---|
| GPT-5 | 52.4% | 55.6% | 0.69 |
| Claude Opus 4 | 44.1% | 37.8% | 0.38 |
| GPT-4o | 25.8% | 26.5% | 0.91 |
| Qwen3-8B | 14.6% | 15.3% | 0.89 |
| Qwen3-14B | 18.3% | 16.7% | 0.77 |
| Qwen3-32B | 18.9% | 18.1% | 0.89 |
| Gemini 2.5 Pro | 34.5% | 31.5% | 0.67 |

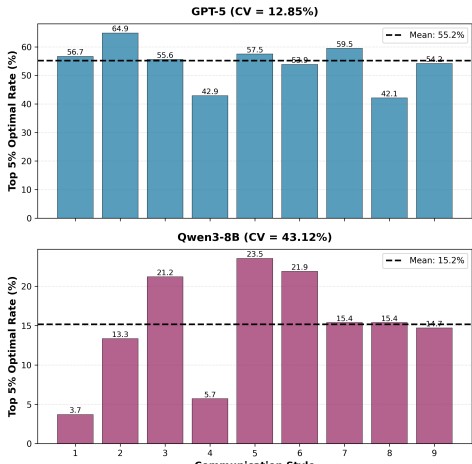

Figure 8: **Example of performance variation across user styles.** We examine model's performance (Top-5% Optima) across different user communication styles (Tab. 10, e.g., polite, rude, casual). To measure whether model achieve similar task performances across different user types, we use Coefficient of Variation (CV): standard deviation across user types normalized by mean. Lower CV indicates higher robustness across user styles. We report CV for different models in Tab. 7

## A.4 AGENTIC WORKFLOW

In our main result (Sec. 4.1), we have focused our benchmark performance on LLM as agents, leveraging their native tool-calling abilities. Recently, many agentic workflows have been developed to further leverage LLM's capabilities without further training. Here, we implement the ReAct (Yao et al., 2022) framework on GPT models.

Our ReAct implementation follows a three-phase iterative cycle: (1) **Think** – the agent explicitly reasons about the current situation, available information, and what tools are needed; (2) **Act** – based on the reasoning, the agent either executes tool calls or provides a final answer; (3) **Observe** – tool execution results are incorporated as observations. This cycle repeats for up to $K$ iterations (we use $K = 10$) until the agent decides to provide a final response.

Concretely, we augment the system prompt with ReAct instructions that require the agent to output structured reasoning in JSON format: {"thought": "detailed analysis", "action": "use_tools" *or* "final_answer"}. In the Think phase, we prompt the model to analyze what information it has, what it needs, and why specific tools would help.

In Table 8, we observe contrasting effects of ReAct across model types. ReAct consistently improves performance on GPT-4.1, with gains of +6.0% in Acceptable Rate and +4.8% in Top-5% Optimal Rate, indicating that adding explicit reasoning before tool-calling enhances the capabilities of non-reasoning models. However, we observe a sharp performance drop for GPT-4o with ReAct enabled

Table 7: **Consistency across communication styles.** Coefficient of Variation (CV) of the Top 5% Optimal Rate across nine user communication styles. GPT-5 exhibits the most stable performance (lowest CV), while open-source models show higher variability. We show model performance breakdown across different user types in Fig. 8.

| Model | CV (%) |
|---|---|
| GPT-5 | 12.85 |
| Claude Opus 4 | 18.90 |
| GPT-4.1 | 26.49 |
| GPT-4o | 28.89 |
| Gemini 2.5 Pro | 29.61 |
| Qwen3-14B | 29.97 |
| Qwen3-32B | 38.75 |
| Qwen3-8B | 43.12 |

Table 8: Performance comparison of GPT models with ReAct (Yao et al., 2022) framework. ReAct consistently improves GPT-4.1 across all metrics but degrades GPT-4o performance due to increased reasoning steps causing context length overflows in 20% of tasks.

| Model | Acceptable Rate | Top-5% Optimal | Top-10% Optimal |
|---|---|---|---|
| GPT-4o | 51.1% | 30.3% | 21.6% |
| GPT-4o + ReAct | 42.8% | 22.1% | 15.3% |
| GPT-4.1 | 65.0% | 37.3% | 25.9% |
| GPT-4.1 + ReAct | 71.0% | 42.1% | 29.4% |

($-8.3\%$ Acceptable Rate). This degradation is primarily due to context length limitations: GPT-4o with ReAct uses significantly more reasoning and tool-calling steps than the baseline. While vanilla GPT-4o encounters no context length overflows, ReAct-enabled GPT-4o exceeds the context limit in 20% of tasks, explaining the large performance drop.

## A.5 ADDITIONAL TOOL-CALL ANALYSIS

In Sec. 6.1, we examined tool-call correctness, search extensiveness, and information gathering quality. Here, we extend the analysis in two directions: (1) tool call efficiency, examining whether models make redundant calls, and (2) detailed breakdown of how information gathering patterns differ across models and relate to task success.

### A.5.1 TOOL CALL EFFICIENCY

We measure tool call efficiency via *duplicate calls*: repeated calls of the same tool with identical inputs within a conversation. While limited repetition can serve to verify tool outputs, excessive duplication signals unproductive reasoning loops. Fig. 9 shows that when models repeat identical calls six or more times, their top-5% success rate drops sharply, suggesting inefficient use of inference compute. Stronger models such as GPT-5 and Claude Sonnet 4 exhibit a balanced pattern: they conduct extensive but non-redundant searches. In contrast, GPT-4o often repeats tool calls excessively, reflecting inefficient iterative behavior that fails to improve outcomes. This analysis complements the extensiveness results in the main paper—it's not just about making many calls, but making *purposeful, varied* calls.

### A.5.2 DETAILED INFORMATION GATHERING ANALYSIS

The main paper showed that gathering more information correlates with task success (Fig. 5). Here, we provide a more detailed breakdown examining how information gathering patterns differ across models and their relationship to task outcomes. In Fig. 10, we visualize the distribution of unique offers discovered per task for GPT-5 and Qwen3-8B, overlaying corresponding success rates within each bin. We observe that Qwen3-8B rarely discovers more than 50 unique offers per task, while GPT-5 frequently explores 100+ offers. When Qwen3-8B does discover a comparable number

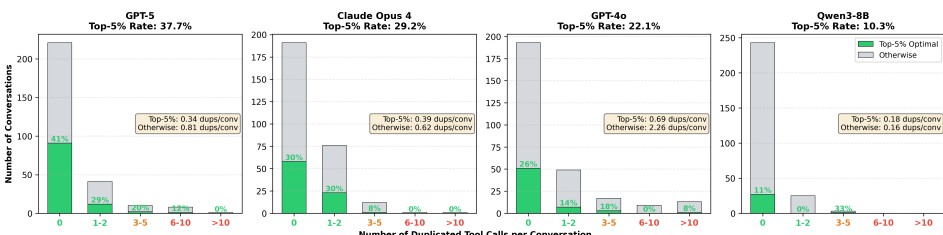

Figure 9: **Tool call efficiency.** Duplicate calls vs. top-5% optimality rate. Excessive duplication (6+ identical calls) sharply reduces task success, indicating redundant reasoning loops.

of offers to GPT-5 (e.g., 20-50 offers), its success rate approaches that of stronger models. This differences explains much of the performance gap where open-source models simply don't gather enough information to reliably identify optimal solutions. After gathering sufficient information, these models also need to reason through hundreds of offers, process and propose a valid itinerary.

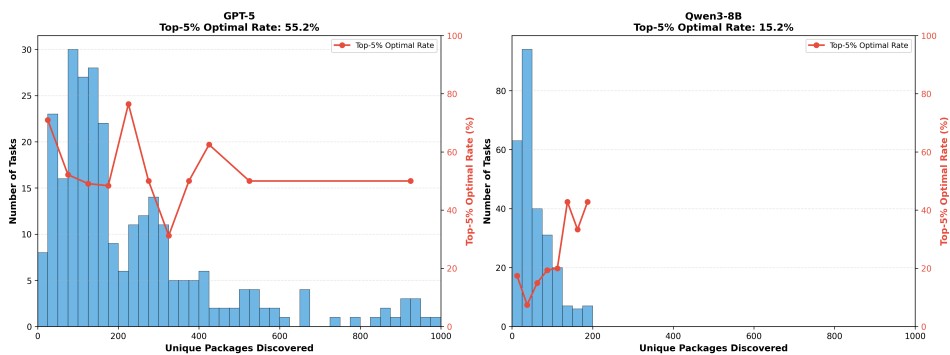

Figure 10: **Information discovery patterns and task success: GPT-5 vs. Qwen3-8B.** For each bin of unique offers discovered, we plot the distribution of tasks and corresponding top-5% optimality rate. Qwen3-8B's performance is largely limited by insufficient exploration (rarely discovering 50+ offers), though it performs comparably to GPT-5 when both discover similar (and small) amounts of information. GPT-5 maintains high success even when reasoning through hundreds of offers.

### A.6 ADDITIONAL ANALYSIS RESULTS

In Fig. 11, we include full results for task level analysis in Sec. 5. In Fig. 12, we include additional result (conversation turn efficiency and acceptable rate) for agent task efficiency analysis in Sec. 4.1.

### A.7 ADDITIONAL CONVERSATION STATISTICS

In Fig. 13 *(top)*, we confirm that the distribution of turns at which users reveal complete task information is identical across all agent models. This validates that the efficiency differences observed in Fig. 3 **(D)** and Fig. 12 arise from agent behavior rather than user variation. Fig. 13 *(bottom)* shows box plots of full conversation lengths, highlighting cross-model differences in dialogue efficiency.

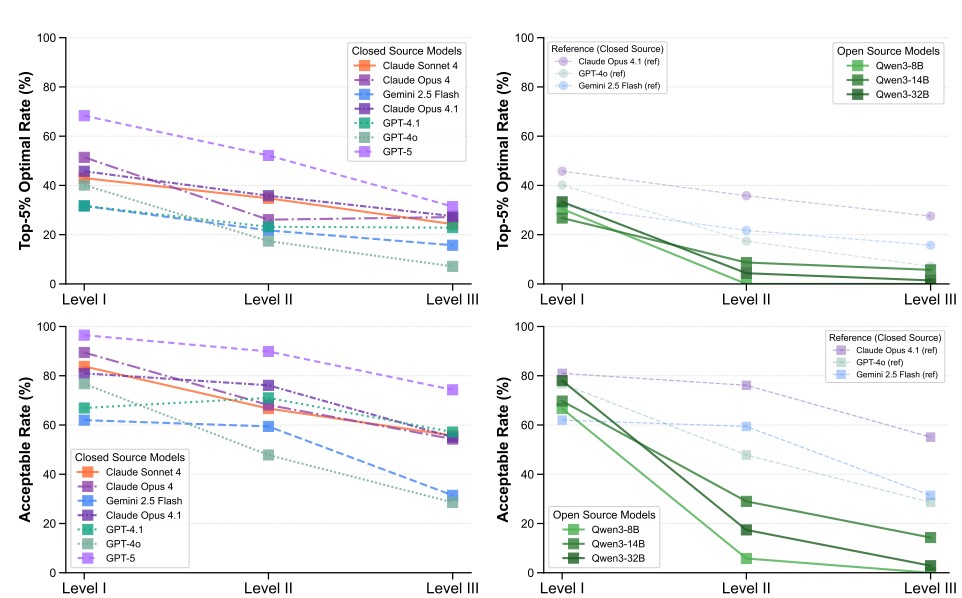

Figure 11: **Task level breakdown performances.** Full result for Fig. 3 **(A)**. As we increase task levels, solving the task requires more complex temporal reasoning and planning. Agents struggles to solve Level II and III tasks and the performance drop is particularly prominent for open-source models.

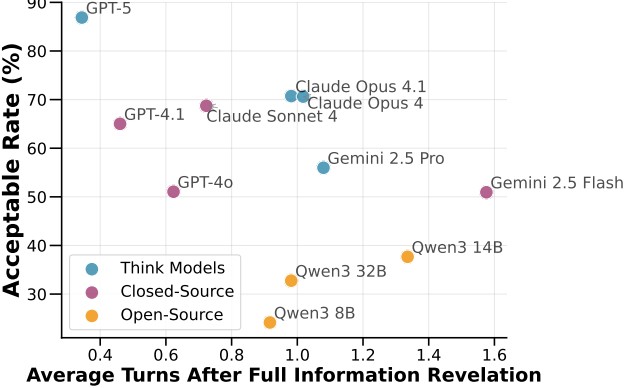

Figure 12: **Conversation efficiency versus Acceptable Rate.** Additional results for Fig. 5 **(D)** In addition to measuring conversation efficiency against optimal rate, we also measure Acceptable Rate.

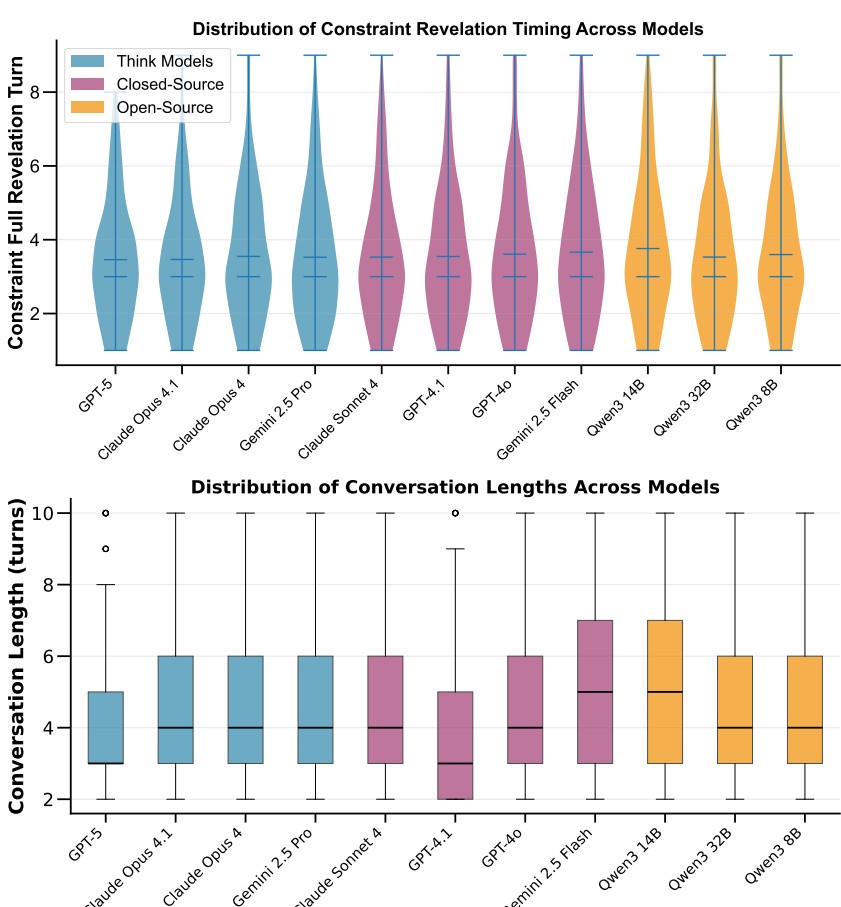

Figure 13: **Conversation efficiency controls.** *Top:* Distribution of user turn $t^*$ when all task information is revealed, showing consistent user behavior across models. *Bottom:* Box plots of overall conversation length, illustrating efficiency differences across agent models. Colors indicate model categories (e.g. open-source, closed-source).

## B    RELATED WORK EXTENDED

**Tool-Use Benchmarks**    Recent work has increasingly recognized the importance of evaluating language model agents in multi-turn settings where information is revealed progressively. Yao et al. (2024) establishes key requirements for realistic agent evaluation, emphasizing the need for agents to interact seamlessly with both humans and APIs over long horizons while adhering to complex policies. However, existing benchmarks (Guo et al., 2024; 2025; Patil et al., 2025; Zhong et al., 2025) often feature simplified instruction-following setups where agents interact autonomously with complete information upfront, lacking realistic human-in-the-loop interaction.Wang et al. (2023) provides multi-turn tool use evaluation but focuses on coding scenarios with overly helpful users, while Patil et al. (2025) introduces multi-step and multi-turn function calling but with predetermined conversation trajectories. Guo et al. (2024) offers comprehensive tool libraries through API crawling, though task instructions based directly on tool calls reduce the challenge of realistic tool selection under uncertainty. Xu et al. (2024) identifies tool selection and usage hallucinations as key failure modes, while Ross et al. (2025) explores when agents should defer tool invocation in favor of clarification dialogue.

**Planning Benchmarks**    Recent planning benchmarks have emerged to evaluate agent capabilities in constraint satisfaction and optimization tasks. Xie et al. (2024) introduces multi-day travel planning with hard constraints and budget optimization, requiring agents to coordinate between multiple APIs for transportation, accommodation, and activities. Valmeekam et al. (2022) provides a comprehensive evaluation framework for planning tasks that require reasoning about state changes and action consequences. Zheng et al. (2024) focuses on realistic planning in natural language with tasks like trip planning, meeting planning, and calendar scheduling, providing tool outputs as context to eliminate tool-use complexity while revealing significant performance drops as problem complexity increases. Kohli et al. (2024) connects compositional and conditional reasoning to flight booking scenarios, presenting detailed user preferences with flight options in multiple-choice format. Zheng et al. (2024) extends planning evaluation to GUI-based environments where agents must navigate complex interfaces to accomplish user goals. These benchmarks emphasize the importance of constraint adherence in realistic planning scenarios. Our work expand on these benchmarks by introduction user preference optimization.

**Task-Oriented Dialogue Systems**    Traditional task-oriented dialogue datasets provide foundational evaluation frameworks that inspire our user simulator design. Budzianowski et al. (2018) and Rastogi et al. (2019) offer crowd-sourced human-to-human dialogues with informable and requestable slots for constraint specification. These datasets encourage goal changes when constraints cannot be satisfied, mimicking real user adaptability. Hosseini-Asl et al. (2020) demonstrates that unified GPT models can handle dialogue state tracking, action prediction, and response generation simultaneously. However, these benchmarks typically lack the tool integration and utility optimization challenges present in modern agent evaluation settings.

**Multi-turn Interaction**    A critical challenge in multi-turn evaluation lies in creating realistic user simulators that can model underspecified preferences and progressive constraint revelation. Laban et al. (2025) demonstrates that current models struggle significantly with synthesizing information across turns, particularly when users reveal requirements in non-sequential "shards" rather than logical order. Abdulhai et al. (2023) offers a toolkit for multi-turn RL evaluation on tasks like city guessing and maze navigation that, despite their simplicity, effectively capture the information-gathering nature of multi-turn conversations. User simulation approaches range from rule-based systems to sophisticated neural models: Lin et al. (2022) uses BERT-based models for generating both semantic actions and natural language utterances, while Shah et al. (2018) employs machine-generated dialogue flows rewritten by humans. Recent preference learning work includes Wan et al. (2025), which introduces turn-based rewards based on belief updates about user types, and Wu et al. (2024); Prabhakar et al. (2025); Zhou et al. (2025), which develops diverse user personas and multi-turn preference datasets for SFT and RL training paradigms.

## C ADDITIONAL TASK DETAILS

### C.1 TASK STATISTICS AND DISTRIBUTION

Our benchmark consists of 241 tasks spanning diverse travel planning scenarios across 20 U.S. National Parks destination . This section provides detailed breakdowns of the task characteristics.

#### C.1.1 CONSTRAINT DISTRIBUTION

Tasks in our benchmark contain both initial (revealed at the first user query) and progressive constraints (revealed during the multi-turn conversation). Fig. 14 shows the distribution of total constraints per task. The majority of tasks have 7-8 constraints. This distribution ensures a balanced mix of task complexity levels, from simpler scenarios with basic requirements to more complex multi-constraint planning challenges.

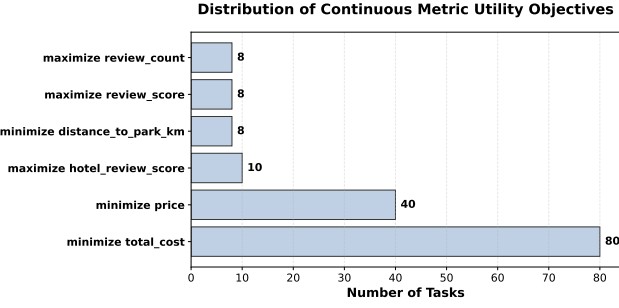

Figure 14: Distribution of total constraints (initial + progressive) across benchmark tasks.

#### C.1.2 TASK TYPE I: SINGULAR METRIC OPTIMIZATION

For tasks using continuous metric optimization, Fig. 15 shows the specific objectives. Price minimization (price for Level II and total cost for level II and III) dominates continuous tasks, reflecting the common real-world constraint of budget optimization. Additionally, we also have review score maximization, distance minimization and review count maximization continuous tasks.

Figure 15: Distribution of optimization objectives for continuous metric tasks.

#### C.1.3 TASK TYPE II: FEATURE COUNT MAXIMIZATION

For feature count maximization tasks, Fig. 16 shows the distribution of target attributes to maximize. The most common scenario involves 8 target attributes, followed by 10 attributes. This distribution creates varying levels of optimization complexity, with tasks requiring agents to balance multiple competing preferences simultaneously.

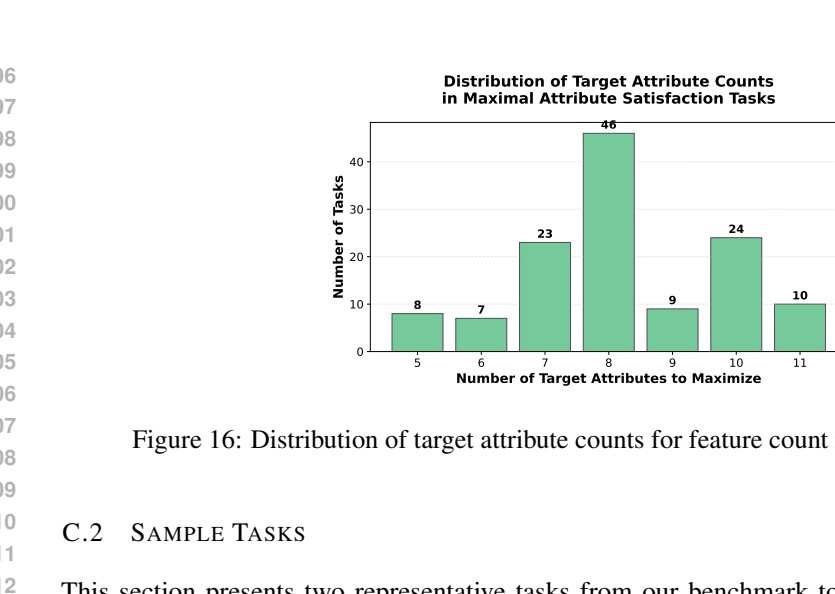

Figure 16: Distribution of target attribute counts for feature count maximization tasks.

## C.2 SAMPLE TASKS

This section presents two representative tasks from our benchmark to illustrate the complexity and diversity of scenarios agents must handle. Each task demonstrates the progressive constraint revelation mechanism and different utility optimization paradigms.

**Task 1: Budget Hiking Trip (Single Metric Optimization) - Level I**

**Initial Query:** "Hi, I'm planning a solo hiking trip to Death Valley National Park and need accommodations for 1 person. I'm looking for a 7-day (6-night) stay sometime in August 2025, but I have no preference on the exact dates. My budget is no more than $1400 for the entire stay. Can you help me find the cheapest options available that meet these requirements?"

**Task Details:**

- **Utility Type:** Single metric optimization (minimize price)
- **Duration:** 7 days (6 nights) in August 2025
- **Location:** Death Valley National Park
- **Budget:** $1,400 maximum

**Initial Constraints:**

- National park: Death Valley National Park
- Guests: 1 person
- Date flexibility: Any 7-day period in August 2025
- Total budget: $\leq$ $1,400

**Progressive Constraints (revealed during conversation):**

- Distance to park: $\leq$ 40 km
- Air conditioning: Required

**Ground Truth Solution:**

- **Hotel:** The Ranch At Death Valley
- **Dates:** August 24-30, 2025 (6 nights)
- **Total Cost:** $1,062.60 ($177.10/night)
- **Utility Score:** 1.0 (optimal price minimization)
- **Feasible Options:** 51 packages within constraints

**Task 2: Business Luxury Retreat (Feature Count Maximization) - Level I**

**Initial Query:** "Hello, I'm planning a 6-day (5-night) business retreat for one person at Grand Teton National Park from August 11th to August 16th, 2025. The company is covering up to $5500 for the trip, so we need to stay within that budget. Could you help me find accommodations that meet these requirements?"

**Task Details:**

- **Utility Type:** Feature count maximization
- **Duration:** 6 days (5 nights), fixed dates
- **Location:** Grand Teton National Park
- **Budget:** $5,500 maximum

**Initial Constraints:**

- National park: Grand Teton National Park
- Guests: 1 person
- Fixed dates: August 11-16, 2025
- Total budget: $\leq$ $5,500

**Target Attributes to Maximize:**

- Gym facilities
- Spa services
- Restaurant on-site
- Airport shuttle
- Star rating $\geq 4.0$
- Review score $\geq 8.5$
- Distance to park $\leq 20$ km
- In-room fridge

**Ground Truth Solution:**

- **Hotel:** The Lodge at Jackson Hole
- **Dates:** August 11-16, 2025 (5 nights)
- **Total Cost:** $3,256.15 ($651.23/night)
- **Utility Score:** 0.75 (6 out of 8 target attributes satisfied)
- **Feasible Options:** 103 packages within constraints

**Task 3: Airport Drive Time Optimizer (Single Metric Optimization) - Level II**

**Initial Query:** "Hi, I'm looking to plan a trip from JFK to Yosemite National Park for one person. I'm completely flexible on dates as long as the trip happens in August 2025. I need a 4-day trip (3 nights hotel) including travel time. It's essential that the hotel is within 1.5 hours from the airport, and I want to keep the total budget for flights and hotel combined under $1,500. Can you help me find the cheapest options meeting these requirements?"

**Task Details:**

- **Utility Type:** Single metric optimization (minimize total cost)
- **Duration:** 4 days (3 nights) in August 2025
- **Location:** Yosemite National Park
- **Budget:** $1,500 maximum for combined flights and hotel

**Initial Constraints:**

- Origin city: JFK
- National park: Yosemite National Park
- Guests: 1 person
- Date flexibility: Any 4-day period in August 2025
- Total budget: $\leq$ $1,500 (flights + hotel)

**Progressive Constraints (revealed during conversation):**

- Hotel to airport drive time: $\leq$ 1.5 hours
- Hotel star rating: $\geq$ 2.5 stars
- Flight must have wifi

**Ground Truth Solution:**

- **Flight:** United Airlines JFK $\rightarrow$ FAT (Fresno)
- **Outbound:** August 26, 2025 at 7:00 AM
- **Return:** August 29, 2025 at 2:00 PM
- **Flight Cost:** $310 (Basic Economy, direct flight with wifi)
- **Hotel:** Yosemite View Lodge
- **Dates:** August 26-29, 2025 (3 nights)
- **Hotel Cost:** $850.50
- **Total Package Cost:** $1,160.50
- **Utility Score:** 1.0 (optimal cost minimization)
- **Feasible Options:** 1,800 packages within constraints

**Task 4: Senior Comfort Tour Maximizer (Feature Count Maximization) - Level III**

**Initial Query:** "Hello, I'm looking to organize a 4-day trip (3 nights hotel) including travel time for two people from LAX to Arches National Park. We want to make sure we include a guided tour at Fiery Furnace. Our schedule is flexible on exact dates, but we want to avoid weekends in August 2025. Our total budget for flights and hotel combined is $6,000. Could you help us arrange this complete travel package?"

**Task Details:**

- **Utility Type:** Feature count maximization
- **Duration:** 4 days (3 nights), weekdays only
- **Location:** Arches National Park
- **Budget:** $6,000 maximum for flights and hotel
- **Special Requirement:** Guided tour at Fiery Furnace

**Initial Constraints:**

- Origin city: LAX
- National park: Arches National Park
- Guests: 2 people
- Date preference: Weekdays only in August 2025
- Total budget: $\leq$ $6,000 (flights + hotel)

- Required guided tour: Fiery Furnace

**Progressive Constraints (revealed during conversation):**

- Outbound departure time: $> 9{:}00$ AM
- Hotel review score: $\geq 8.0$

**Target Attributes to Maximize:**

- Hotel has restaurant on-site
- Hotel has air conditioning
- Hotel star rating $\geq 3.0$
- Distance to park $\leq 10$ km
- Direct flight preferred
- Seat selection included
- In-flight wifi available

**Ground Truth Solution:**

- **Flight:** United Airlines LAX $\rightarrow$ CNY (Moab)
- **Outbound:** August 4, 2025 at 4:00 PM
- **Return:** August 7, 2025 at 8:00 AM
- **Flight Cost:** $560 (Economy, direct flight with wifi)
- **Hotel:** Field Station Moab (4-star)
- **Dates:** August 4-7, 2025 (3 nights)
- **Hotel Cost:** $870.30
- **Permit:** Fiery Furnace Guided Tour on August 6, 2025
- **Total Package Cost:** $1,445.30
- **Utility Score:** 0.71 (5 out of 7 attributes satisfied)
- **Feasible Options:** 19,589 packages within constraints

## D  AVAILABLE TOOL SCHEMA

Agents have access to four categories of tools for comprehensive travel planning and utility maximization. The tool schema is defined using OpenAI function calling format with JSON schema specifications.

### D.1  AIRPORT AND FLIGHT TOOLS

**search_airports**
Description: Find airports serving a specific national park with driving times
Parameters: park_name - National park name (e.g., 'Yosemite', 'Grand Canyon')
Returns: List of airports with airport_code, airport_name, city, drive_hours, airport_type

**search_flights**
Description: Search for round-trip flights between airports on specific dates
Parameters: origin (3-letter airport code), destination (3-letter airport code), departure_date (YYYY-MM-DD), return_date (YYYY-MM-DD), passengers (opt.), max_price (opt.), airline_preference (opt.), booking_class (opt.), direct_flights_only (opt.)
Returns: Flight packages with pricing, schedules, airline info, policies, package_id

**get_flight_details**
Description: Get detailed information about a specific flight package
Parameters: package_id - Flight package ID (format: flight_templateid_depday_retday)
Returns: Complete flight details including route, schedule, dates, airline, policies, pricing

**get_airport_coordinates**
Description: Get latitude/longitude coordinates for an airport
Parameters: airport_code - 3-letter airport code (e.g., 'LAX', 'COD')
Returns: Airport coordinates (airport_code, name, latitude, longitude)

### D.2  PERMIT TOOLS

**list_permits**
Description: List all available park permits and guided tours for a national park
Parameters: park - National park name (e.g., 'Yosemite National Park')
Returns: Available permit types (day_hikes, guided_tours, backpacking) with template_name

**search_permit_availability**
Description: Search park permit availability for specific permit template
Parameters: park (park name), template_name (exact from list_permits), date_range (opt.), duration_days (opt. for backpacking)
Returns: Available dates with package_ids for booking permits

### D.3  ACCOMMODATION TOOLS

**AccommodationSearch**
Description: Search for available accommodations by location and date range
Parameters: location, start_date, end_date, num_guests, min_price_nightly (opt.), max_price_nightly (opt.), min_review_score (opt.), free_cancellation (opt.), no_prepayment_needed (opt.), has_parking (opt.), breakfast_included (opt.), is_pet_friendly (opt.)
Returns: Hotels with group booking packages, including package_id, pricing, policies

**get_hotel_details**
Description: Get detailed hotel information. Supports fuzzy matching
Parameters: hotel_identifier - Hotel ID or name (supports partial names)
Returns: Hotel details including amenities, location, star_rating, review_score

**get_room_details**
Description: Get detailed room information including bed configuration
Parameters: package_id - Package ID (format: pkg_configid_startday_endday_rooms)
Returns: Room details including room_type, beds, amenities, policies

**search_location_name**
Description: Search for location names using fuzzy matching
Parameters: query - Location search query (can be partial/informal)
Returns: Matching locations with confidence_score and location_type

## D.4 UTILITY TOOLS

**get_weekday**
Description: Returns the weekday name for a given date
Parameters: date_str - Date in YYYY-MM-DD format
Returns: Weekday name (e.g., 'Monday')

**calculate**
Description: Evaluate a basic arithmetic expression
Parameters: expression - Arithmetic expression (e.g., '100 + 200 * 3')
Returns: Numerical result

**calculate_distance**
Description: Estimate driving time between two locations using coordinates
Parameters: lat1, lon1 (starting point), lat2, lon2 (destination)
Returns: Estimated driving time (e.g., '2h 15m') with distance

**Notebook**
Description: Agent's scratch pad for notes and memory
Parameters: action ('write', 'read', 'delete', 'list_all'), input_data (for 'write'), index (for 'read'/'delete')
Returns: Confirmation and content based on action

## D.5 VALIDATION TOOLS

**recommend_hotel**
Description: MANDATORY validation for hotel package IDs before recommendation
Parameters: package_ids (list of 1-3 IDs), reasoning (explanation)
Returns: Validation results confirming package ID validity

**recommend_flight**
Description: MANDATORY validation for flight package IDs before recommendation
Parameters: flight_ids (list of 1-3 IDs), reasoning (explanation)
Returns: Validation results confirming package ID validity

**recommend_permit**
Description: MANDATORY validation for permit package IDs before recommendation
Parameters: permit_ids (list of 1-3 IDs), reasoning (explanation)
Returns: Validation results confirming package ID validity

# E LLM USER SIMULATOR

## E.1 USER SIMULATOR DESIGN AXIS

As introduced in Sec. 3.4, our user simulator varies along four independent design axes to capture diverse conversational dynamics. These axes are summarized in Table 9 and detailed below.

Table 9: **User Simulator Design Axes.** The four axes are independent, yielding $3 \times 2 \times 2 \times 9 = 108$ possible user simulator types.

| Design Axis | Parameter Values | Description |
|---|---|---|
| **Progressive Constraint Revelation** | • Slow (33.3%)
• Medium (33.3%)
• Fast (33.3%) | Controls how quickly users reveal hard constraints during the conversation. Each user is initialized as slow (0.4–0.6), medium (0.6–0.8), or fast (0.8–1.0) in revelation probability, and continues revealing constraints until all have been disclosed. |
| **Trust and Communication Patterns** | • Trusting (80%)
• Questioning (20%) | Determines whether users accept or challenge the agent's recommendations once all constraints are revealed. Trusting users accept the output, while questioning users ask the agent to verify its solution. |
| **Constraint-Checking Reliability** | • Reliable (50%)
• Unreliable (50%) | Models users' attentiveness to constraint violations. Reliable users notice and point out violations in the agent's suggestions based on revealed task information, whereas unreliable users often overlook them. |
| **Conversation Style** | 9 distinct styles (see Tab. 10) | Captures variation in user tone and personality (e.g., polite, rude, agitated). Style affects conversational tone only and does not alter the factual content of the dialogue. |

**Progressive constraint revelation.** This axis controls how quickly users disclose their hard constraints throughout the conversation. At initialization, each user is randomly assigned to one of three revelation-speed groups with equal probability: *slow* (0.4–0.6), *medium* (0.6–0.8), or *fast* (0.8–1.0). At each turn, users reveal new constraints according to their assigned rate until all constraints have been stated.

**Trust and communication patterns.** After all constraints are revealed, users differ in whether they accept or question the agent's recommendations. Each user is initialized as *trusting* (80%)—accepting the agent's suggestions—or *questioning* (20%)—requesting the agent to verify its outputs.

**Constraint-checking reliability.** This axis models users' attentiveness to constraint violations. *Reliable* users (50%) identify and flag violations in the agent's recommendations based on revealed task information, whereas *unreliable* users (50%) often fail to notice such issues.

**Conversation style.** Finally, we vary users' communication tone and personality by sampling one of nine distinct conversation styles (e.g., polite, rude, agitated) at initialization. These styles affect tone only and do not alter the semantic correctness of the dialogue.

## E.2 DYNAMIC PROMPTING

Building on the four design axes described in App. E.1, we implement a dynamic prompting framework that governs how user simulators interact with the LLM agent across turns. This mechanism ensures that conversational behaviors, such as when to reveal new constraints or whether to question the agent, emerge in a controlled yet diverse manner.

Fig. 17 **(B)** illustrates the dynamic prompting structure, and App. E.3 provides the verbatim prompt template. At each dialogue turn, dynamic fields are updated to reflect the current conversation state—what task information has been revealed, what remains hidden, the user's feedback on the agent's last response, and whether questioning behavior should be triggered (e.g., asking the agent to double-check a recommendation). Static fields, by contrast, encode the user persona and task-level context.

Table 10: 9 Conversation Styles in User Simulator

| # | Description |
|---|---|
| 1 | You are extremely polite and want to be personal with the agent. You thank them frequently, ask how their day is going, and share small personal details about your trip or life. You use phrases like 'I really appreciate this' and 'you're being so helpful' constantly. |
| 2 | You are overly apologetic and worry about being a burden. You constantly say 'I'm sorry to bother you' and 'I hope this isn't too much trouble' and 'I know you're busy but...' even when asking perfectly reasonable questions. |
| 3 | You are extremely suspicious and paranoid about being scammed. You question everything with 'are you sure about that?', and 'why should I trust this recommendation?' And you always ask the agent to show reasoning and thinking traces. |
| 4 | You constantly demand that the agent explain their reasoning and justify their recommendations. You say things like 'Show me your thinking process', 'How did you arrive at that conclusion?', 'Prove to me why this is the best option', and 'I need to understand your logic before I can trust this recommendation'. You question every suggestion with 'But why?' and 'What's your reasoning behind this?' |
| 5 | You are extremely verbose and provide way too much background information. You tell long stories about why you're traveling, what happened last time, your family history, and go off on tangents before getting to your actual point. |
| 6 | You are scattered and repetitive in your communication. You repeat the same points multiple times within or across conversation terms. |
| 7 | You are very dramatic and emotional in your language. Everything is either 'amazing' or 'terrible', 'perfect' or 'a complete disaster'. You use lots of exclamation points and phrases like 'this is so exciting!' or 'oh no, that won't work at all!' |
| 8 | You get aggressive and agitated very easily. You start the conversation very patient but as soon as the agent doesn't get what you want right away, you use swear words and you use a lot of threatening expressions. |
| 9 | You are very direct, organized, and businesslike in your communication. You structure your requirements using bullet points or numbered lists, communicate everything logically and clearly, and don't waste time on pleasantries. You say things like 'My requirements are: 1) X, 2) Y, 3) Z' and 'Please provide options that meet these criteria.' You prefer structured, logical responses and often summarize key points. |

Together, this dynamic prompting design enables diverse yet reproducible user behaviors, allowing systematic evaluation of how LLM agents adapt to different interaction styles and reasoning challenges.

### E.3   SIMULATOR USER PROMPT

Below we list the dynamic prompt template we used for the LLM user simulator. The blue fields are fields are populated at each conversation turn based on user persona and the conversation state.

---

CORE INSTRUCTIONS

**Role Definition:**
*You are the **user** (not the assistant) responding to a travel assistant's recommendation. Your goal is to find one complete itinerary (flights + hotel + optional permits) that satisfies your hard constraints (non-negotiable) while expressing soft preferences as utility objectives.*

---

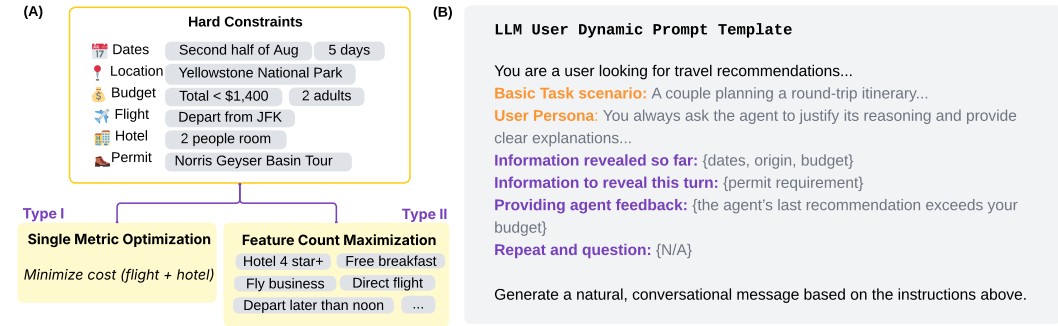

Figure 17: **Examples of task types and dynamic user simulator prompt.** **(A)** Two task types are defined based on soft preference-optimization objectives. Each task type includes hard constraints but differs in its optimization goal (Sec. 3.2). **(B)** The dynamic LLM user simulator prompt (Sec. 3.4) governs multi-turn conversation dynamics. The system prompt combines static instructions (orange), fixed throughout the dialogue, with dynamic fields (purple), which are updated at each turn based on the evolving conversation state.

**Conversation Context:**
*Your conversation style, attention level, and persona are specified below and remain consistent throughout.*
*You MUST consistently embody your communication style—your personality should be clearly evident in every response.*
*Dynamic fields update each turn with constraint check results, utility instructions, and behavioral triggers.*

DYNAMIC PROMPT FIELDS

- **Conversation Style:** {persona_description}
- **Attention Level:** {attention_level}
  *High—carefully reviews all details;*
  *Medium—notices key points but may miss minor ones;*
  *Low—focuses on big picture, often misses specific violations.*
- **Hard Constraints (non-negotiable):**
  {hard_constraints}
- **Utility Objective (soft preference):**
  {utility_objective}
  *Note: DO NOT check or comment on recommendations based on utility (soft preferences).*
- **Hard Constraint Check Results:**
  {constraint_check_natural_response}
- **Utility Instruction:**
  {utility_instruction}
- **Adding Hard Constraint Instruction:**
  {constraint_instruction}
- **Questioning & Verification Instruction:**
  {questioning_instruction}

RESPONSE COMPONENTS

*Your response has five components:*

1. Respond to agent questions if asked
2. Incorporate hard constraint check results
3. State utility objective if you haven't done so in the conversation
4. Mention additional new hard constraints if instructed

5. Repeat utility goals/constraints if instructed

**Important Guidelines:**

- You only mention constraint violations listed in HARD CONSTRAINT CHECK RESULTS. DO NOT independently check recommendations against your constraints.
- Answer agent questions using ONLY information explicitly stated in your constraints and utility objective. Ignore questions that cannot be answered from these sources. DO NOT hallucinate information.
- Follow all additional hard constraint injection instructions exactly.
- Be authentic to your persona and natural in conversation. Do not repeat information already mentioned unless instructed to be verbose and repetitive.
- Do not try to be helpful or suggest what the agent should do—expect the agent to do the work.
- Your goal is to walk away with one itinerary recommendation. Do not ask the agent to list multiple itineraries.
- Do not ask the agent to lock-in or book anything. Express satisfaction (e.g., "this looks pretty good") or dissatisfaction (when agent violates constraints or doesn't provide a recommendation), but do not request booking.

RESPONSE ANALYSIS WORKFLOW

**Step 1: Analysis**
Generate an analysis covering ALL 5 aspects in order:

1. **question_response:** Identify if agent asked questions answerable from constraints/utility
2. **constraint_check:** Review constraint check instructions; decide if violations need mentioning
3. **state_utility:** Check if you need to state/re-state your utility objective
4. **reveal_progressive_constraint:** Check if instructed to reveal new progressive constraints
5. **question_and_verify:** Check if instructed to question and verify the recommendation

**Step 2: Response**
Generate a natural response continuing the conversation as a user. Be authentic to your persona and respond naturally according to conversation flow. Do not repeat what was already stated unless instructed to be repetitive.

RESPONSE EXAMPLES

```
{
  "analysis": "1. question_response: Agent not asking questions, skip.
            2. constraint_check: Constraint check indicates flight
                time violation, I will mention that.
            3. state_utility: Already stated utility objective, skip.
            4. reveal_progressive_constraint: Not instructed to reveal
                new constraints, skip.
            5. question_and_verify: Not instructed to question, skip
                .",

  "user_response": "Thank you for finding this travel package! I
      really appreciate the effort. However, I noticed the outbound
      flight departs at 3:30 PM, but I specifically need a morning
      departure before noon. Could you find an earlier flight option
      that works with the rest of the package?",
}
```

OUTPUT FORMAT

Respond with JSON:

```
{
  "analysis": "Structure your analysis covering all 5 aspects:
                1. question_response: ...
                2. constraint_check: ...
                3. state_utility: ...
                4. reveal_progressive_constraint: ...
                5. question_and_verify: ...",
  "user_response": "what user would naturally respond based on
      analysis",
  "terminating_condition": "continue"
}
```

# F DATABASE DETAILS

## F.1 CONSTRUCTION DETAILS

We constructed the accommodation database through a systematic data collection pipeline using the Booking.com API (accessed via RapidAPI). For each target destination (U.S. National Parks), we performed hotel discovery within the park boundary for a fixed date range (August 2025), retrieving up to 20 properties per location. For each property, we collected two categories of information: (1) static hotel details, including metadata, geographic coordinates, guest reviews, and facility amenities; and (2) dynamic availability data, capturing daily room inventory, prices, and booking conditions. To record temporal variation, we queried single-night stay windows across the entire month and stored all raw API responses as JSON files.

We then normalized the raw data into a relational SQL database consisting of three interconnected tables: *hotels*, *rooms*, and *offers*. The *hotels* table stores property-level attributes such as star rating, review score, distance to the nearest park, brand affiliation, and binary amenity flags (e.g., pool, restaurant, parking). The *rooms* table captures room-level characteristics, including bed configuration, occupancy limits, and in-room amenities (e.g., kitchenette, air conditioning), with foreign-key linkage to the parent hotel. The *offers* table contains time-varying booking data at daily granularity, recording prices, availability counts, cancellation policies, meal plans, and prepayment requirements.

We constructed the flight database using a similar approach. We selected ten major cities as departure locations and, for each national park destination, included both nearby regional and large city airports. The database covers four major airlines and two flight types: direct and one-stop. We organized the data into two SQL tables: a *flights* table containing static flight attributes (flight number, origin, destination, duration, and onboard amenities such as Wi-Fi and pet policy), and a *prices* table capturing daily flight availability and pricing dynamics.

## F.2 DATABASE OVERVIEW

Fig. 18 summarizes the accommodation database. We selected up to 20 hotels for each national park, except in a few cases where limited availability was observed for August 2025. The figure also reports the distribution of daily prices, ratings, and distances to park entrances. These statistics confirm that the dataset captures realistic diversity across destinations. For instance, Rocky Mountain National Park exhibits substantially higher average prices than other parks (top right), while Death Valley and Everglades National Parks have few nearby lodging options (bottom right).

Fig. 19 presents analogous statistics for the flight database. Flight prices follow a long-tailed distribution, reflecting real-world market variability. Regional airports generally have fewer available routes compared to large city airports, consistent with realistic flight coverage patterns.

Our realistic database captures nuanced temporal and spatial patterns often missing from synthetic datasets. Two notable examples are shown in Fig. 20. **Cyclical pricing trends:** Average prices rise sharply during weekends (Friday–Sunday) when room availability decreases, showing a strong short-term anti-correlation between price and supply. Interestingly, broader price trends across late August remain elevated but do not exhibit this anti-correlation. **Hotel star distribution:** Examining the top 20 recommended hotels[2] for each park reveals that not all destinations cater to luxury travelers, with limited 4-star and above properties in several locations.

---

[2]As ranked by `Booking.com`.

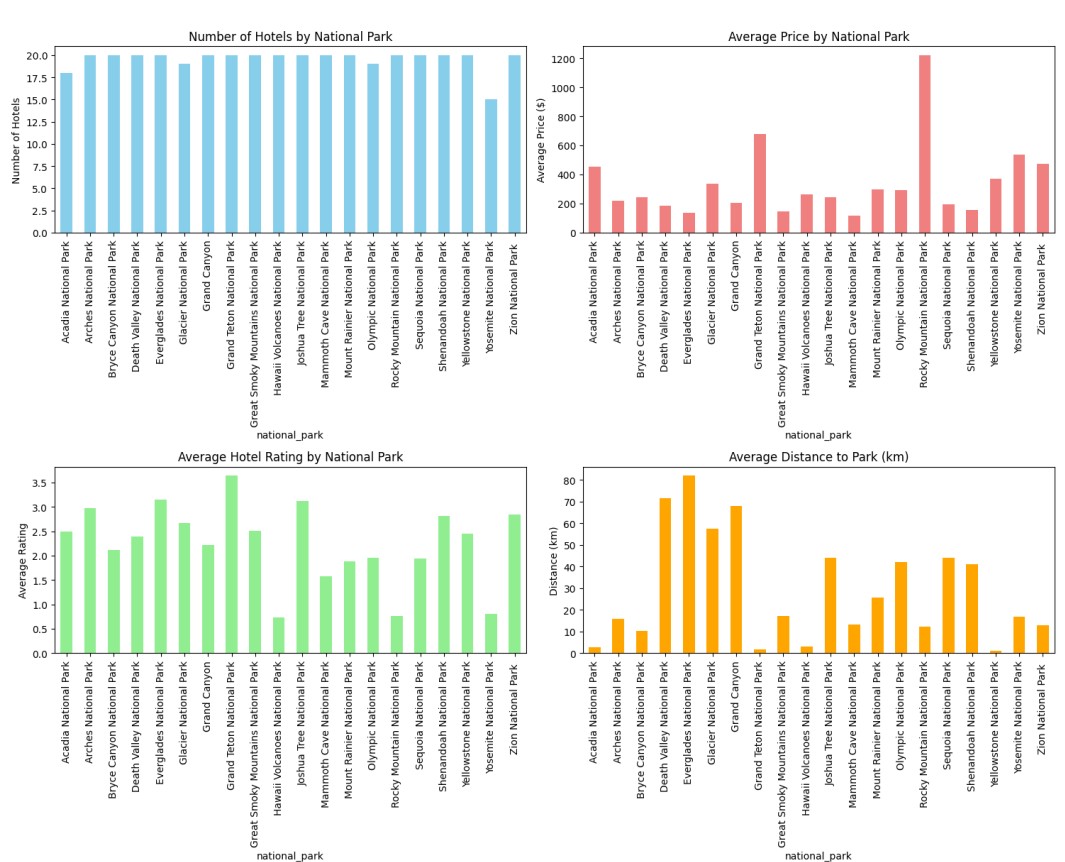

Figure 18: **Hotel database overview.** Summary statistics of the accommodation database. *Top left:* Number of hotels per national park. *Top right:* Average daily room prices by park. *Bottom left:* Average guest ratings. *Bottom right:* Average distance of hotels to the park center.

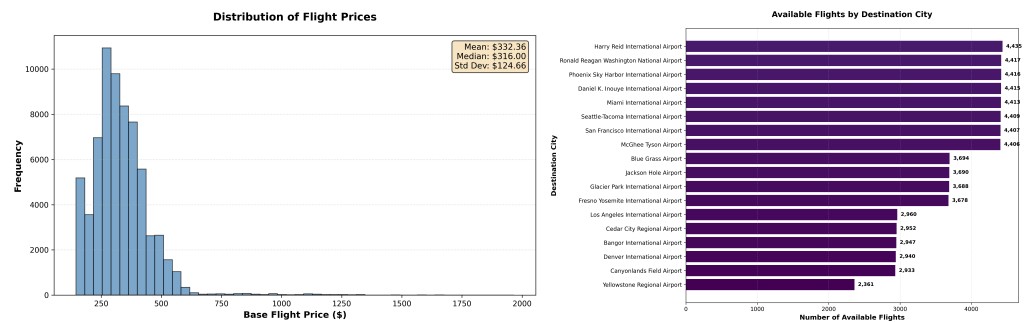

Figure 19: **Flight database overview.** *Left:* Distribution of flight prices across all routes. *Right:* Number of available flights per destination airport.

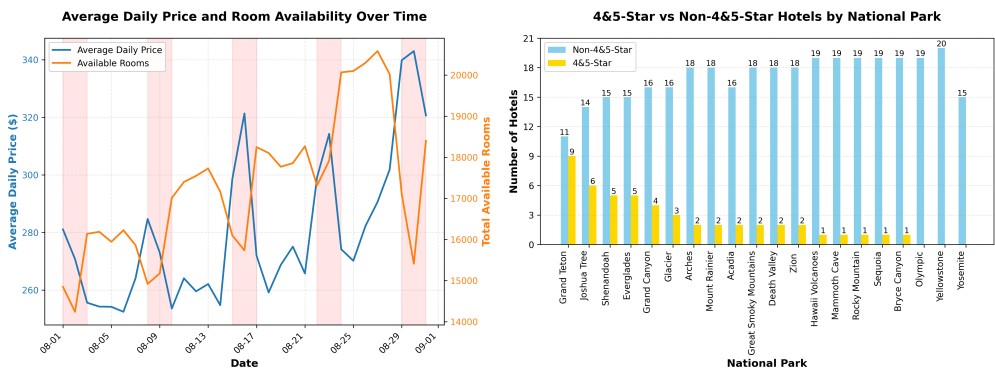

Figure 20: **Real-world patterns in the accommodation database.** *Left:* Relationship between daily price and room availability. Weekend periods (shaded) show higher prices and lower availability, indicating strong short-term price–demand coupling. Longer-term price shifts across August show no such anti-correlation. *Right:* Fraction of high-end (4-star and above) hotels among the top 20 recommended properties per destination, illustrating variability in luxury travel options.

## G EXPERIMENT DETAILS

### G.1 AGENT SYSTEM PROMPT

CORE INSTRUCTIONS

**Role Definition:**
*You are a helpful and proactive travel planning assistant. The current date is June 1st, 2025. Your goal is to help the user find the best travel itinerary (flights + hotel + optional permits) by satisfying their constraints and maximizing their implicit preferences.*

**Critical Requirement:**
*You MUST respond ONLY in valid JSON format using the exact schema below. Do NOT include any text outside the JSON structure.*

AGENT WORKFLOW

1. **Engage in conversation:** Natural, friendly interaction to understand user needs for complete travel packages
2. **Use relevant tools:** Find available flights, hotels, and permits based on user criteria
3. **Validate recommendations:** MUST use `recommend_itinerary` tool to validate complete package before response
   - Provide flight package IDs, hotel package IDs, and optional permit IDs
   - Explain reasoning for the complete itinerary selection
4. **Optional note-taking:** Use `Notebook` tool as scratch pad for complex itinerary planning

RESPONSE FORMAT SPECIFICATION

**When making an itinerary recommendation (after validation):**

```
{
  "message": "I found a great travel package for your Yosemite trip!

            Flight Package:
            - United Airlines: JFK - SFO
            - Outbound: Aug 15 at 10:30 AM
            - Return: Aug 19 at 6:00 PM
            - Cost: $450

            Hotel Package:
            - Yosemite View Lodge (3-star)
            - 4 nights: Aug 15-19
            - Cost: $1,200

            Total Package: $1,650",
  "formal_recommendation": {
    "flight_package_ids": ["flight_pkg_UA123_JFK_SFO_0815"],
    "hotel_package_ids": ["hotel_pkg_YVL_0815_0819_2"],
    "permit_ids": [],
    "reasoning": "This itinerary offers morning departure as
        requested,
                stays within budget, and provides convenient access
                to Yosemite National Park."
  }
}
```

**When NOT making a recommendation:**

```
{
```

```
  "message": "You are welcome! Hope you have a wonderful trip!",
  "formal_recommendation": "None"
}
```

IMPORTANT REQUIREMENTS

- Always include both `"message"` and `"formal_recommendation"` fields in JSON response
- User will only see the `"message"` field—include all necessary travel details there
- Response must be ONLY the JSON object—no text before or after the JSON structure
- The entire response must be parseable as valid JSON
- For itinerary-level recommendations, always recommend ONE complete itinerary (not multiple options)

This prompt design ensures that agents provide consistent, evaluable responses for complete travel itineraries while maintaining natural conversational flow. The mandatory JSON format with formal recommendations enables precise evaluation against ground truth solutions for flights, hotels, and permits as integrated packages.

## G.2 AGENT LLM CONFIGURATION AND SAMPLING

We employ diverse model configurations to capture a broad spectrum of agent capabilities:

**Agent Model Settings:** Different models require specialized configurations. GPT-5 agents use the default thinking setting (medium level thinking) to leverage their enhanced reasoning capabilities. Qwen3 models employ sampling temperature 0.7 as recommended by their documentation, with thinking mode enabled to improve multi-step reasoning performance. All other agent models use temperature 0 (greedy decoding) to ensure deterministic and reproducible results across evaluation runs.

## G.3 AGENT RESPONSE FORMAT VALIDATION AND ANSWER EXTRACTION

Package IDs serve as unique identifiers that enable deterministic mapping between agent recommendations and ground truth solutions. Each package ID encodes the complete booking configuration (hotel, room type, dates, occupancy, flight number, booking class) ensuring that evaluation is based on exact matches rather than fuzzy similarity metrics.

To ensure precise evaluation alignment with ground truth solutions, agents must respond in a structured JSON format containing both a natural language message visible to users and a formal recommendation section with specific package IDs. This dual-format approach maintains conversational naturalness while enabling exact matching against benchmark solutions.

We also provide agent format validation tool, which serves as a mandatory validation checkpoint, preventing agents from hallucinating non-existent package IDs. This tool verifies that all recommended packages exist in the accommodation database before agents include them in their final recommendations, substantially reducing evaluation noise from invalid responses. For hotel recommendation (Level I), if agent recommends multiple options, we pick the best one for evaluation. For itinerary-level recommendation (Level II and III), due to complexity, we ask the agent to always recommend for one valid itinerary.

## H  HUMAN EVALUATION

### H.1  HUMAN EVALUATION INSTRUCTION

We provide the complete annotation protocol and instructions used for human evaluation of the LLM user simulator quality in Section 4.2.

**Annotation Task Overview**

Human annotators evaluated 198 user responses sampled from 45 full conversations. Each response was assessed for both objective errors and subjective quality metrics by third-party independent expert annotation services.

**Task Context for Annotators**

**Understanding the Travel Planning Task**

**Hard Constraints (Must-Have Requirements):** Non-negotiable requirements forming a checklist where EVERY item must be satisfied. Examples include:

- "Must allow pets" - if not pet-friendly, the hotel is unacceptable
- "Must accommodate at least 4 people" - 3 people max would fail this requirement
- "Must be within $800 total" - anything over $800 fails

**Optimization Objective:** Once all must-haves are met, this determines which option to choose:

- "I want the cheapest option" - among all acceptable hotels, pick the lowest price
- "I want the highest rated" - among all acceptable hotels, pick the best reviews
- "I want the most amenities from my wish-list" - among all acceptable hotels, pick the one with most desired features

**User Simulator Configuration**

**Persona Attributes** Each simulated user follows a defined persona with three key attributes that annotators must consider:

- **trust_level**: "suspicious" or "trusting" - Whether the user blindly trusts the agent or asks for reasoning and double-checking
- **attention_level**: "low", "medium" or "high" - Whether the user pays attention to agent's response and notices obvious mistakes (such as hard constraint violations)
- **communication_style**: Verbal style defining formality and expression patterns

**User Script Components** Annotators were provided with the following script information for each user:

- **Hard constraints**: Non-negotiable requirements that MUST be satisfied
- **Utility objective**: What makes one option "best" once constraints are met
- **Communication style**: How the user expresses themselves based on their personality

**Core User Responsibilities**

Annotators evaluated whether users fulfilled six core responsibilities at each turn:

1. **State what makes a hotel "best" for them**: User clearly explains their optimization goal
   - Example: "I want the cheapest option that meets all my requirements"
   - Example: "I'm looking for the highest-rated place that fits my needs"
2. **Reveal must-have requirements**: User shares new non-negotiable requirements
   - Example: "Oh, I also need air conditioning - that's essential for me"

- Example: "I forgot to mention, it absolutely must have free parking"

3. **Verify recommendations meet ALL requirements**: User checks if suggested hotels satisfy every must-have

   - Example: "Wait, that costs $850 but I said my budget is maximum $800"
   - Example: "Does this hotel allow pets? That's a must-have for me"

4. **Answer agent's clarifying questions**: When agent needs more information, user provides it based on their script

5. **Restate requirements if needed**: Suspicious users may repeat their needs to ensure agent understood

   - Example: "Just to be clear, it MUST be pet-friendly AND under $800 total"

6. **Request formal recommendations**: If agent doesn't properly flag their recommendations, user asks them to formally recommend options

*Important Note for Annotators:* Users should not invent information beyond what's provided in their script.

**Error Detection Categories**

Annotators tagged each response for six types of objective errors (True if error detected, False otherwise):

| No. | Error Type | Description | Example |
|-----|-----------|-------------|---------|
| 1 | Factual Hallucination | User mentions requirements or preferences not in their script, changing the task | User adds "I need a gym" when script doesn't mention this |
| 2 | Revelation Instruction Following Failure | User was told to reveal a new must-have requirement but didn't | Instruction says "reveal need for parking" but user doesn't mention it |
| 3 | Revelation Accuracy Failure | User mentioned a requirement but unclearly, making it seem optional instead of must-have | Says "parking would be nice" instead of "I need parking" (must-have) |
| 4 | Recommendation Check Failure | User incorrectly claims a hotel violates their requirements when it actually meets them | Says "this is over budget" when hotel is actually within budget |
| 5 | Repetition Failure | User was told to restate requirements but didn't | Instruction says "repeat your budget constraint" but user doesn't |
| 6 | Critical Comments | Any other significant errors affecting the task | Note any issues not covered above |

**Quality Assessment Rubrics**

**Clarity (1-5 Scale)**

*Evaluation Question:* Has the user clearly communicated:

- What their must-have requirements are (and that they're non-negotiable)?
- What makes one hotel "better" than another for them?
- Any problems with the agent's recommendations?

*Evaluation Goals:*

- User can have different communication styles (casual, formal, etc.) but the core message must be clear
- The agent should understand both the requirements AND the optimization goal

| Score | Quality Level | Description | Example |
|---|---|---|---|
| 5 | Excellent | Crystal clear communication of requirements and goals | "I need a pet-friendly hotel under $500 total. Among options meeting these requirements, I want the cheapest." |
| 4 | Good | Clear communication with no confusion about the task | Requirements and goals are clear, minor wording issues don't affect understanding |
| 3 | Acceptable | Mostly clear but some minor ambiguity that won't affect results | Says "I prefer under $500" when they mean it's required, but context makes it clear |
| 2 | Poor | Unclear communication that could lead agent to wrong recommendations | Mixes up must-haves with nice-to-haves, unclear about optimization goal |
| 1 | Critical | So unclear the agent will likely fail the task due to miscommunication | Contradictory requirements, no clear goal, or completely confusing instructions |

**Contextual Appropriateness (1-5 Scale)**

*Evaluation Question:* Does the response logically and naturally follow the conversation so far?

*Evaluation Goals:*

- User maintains natural conversation flow
- User demonstrates awareness of entire conversation history

| Score | Quality Level | Description |
|---|---|---|
| 5 | Excellent | As good as you would write or better |
| 4 | Good | Good without any obvious issues |
| 3 | Acceptable | A bit unnatural but overall harmless to the conversation flow |
| 2 | Poor | Noticeable issues that do not follow the conversation logic at all |
| 1 | Critical | Response is completely disconnected from the conversation logic (e.g., User starts acting as an agent) |

**Annotation Workflow**

**Process for Each Turn** For each user response (excluding turn 0), annotators:

1. Compare the given user response to the instructions given to the user
2. Answer 6 true/false questions to identify any errors that occurred
   - Answer **True** if error or failure is identified
   - Answer **False** if no error or if question is not applicable
3. Answer 2 rubric grading questions to score the quality of the user's response

## H.2 ADDITIONAL HUMAN EVALUATION RESULTS

In Sec. 4.2 (Tab. 2), we presented user simulator validation results aggregated over all user types (App. E.1). As described in App. E.1 and Tab. 9, the simulator design varies along four main axes: (1) progressive constraint revelation, (2) trust and communication patterns, (3) constraint-checking reliability, and (4) conversation style. Tables 11, 12, and 13 report results broken down by the first three axes. We omit the breakdown by conversation style, as each of the nine styles has too few samples and primarily affects tone (e.g., polite vs. rude) rather than the factual content or correctness of the interaction. Overall, while we observe minor variations in simulator quality across user types, the consistently high scores indicate that simulator quality is unlikely to be the main cause of task failures.

Table 11: Human evaluation of user simulator quality. Results breakdown by design axis (1) Progressive Constraint Revelation (Tab. 9).

| Metric | Slow (n=72) | Medium (n=75) | High (n=19) |
|---|---|---|---|
| **Quality Scores (1–5) ↑** | | | |
| **Clarity** | $3.83 \pm 1.12$ | $3.96 \pm 1.14$ | $3.95 \pm 1.10$ |
| Median | 4.00 | 4.00 | 4.00 |
| Percentiles (25/50/75/90/95) | 3 / 4 / 5 / 5 / 5 | 3.5 / 4 / 5 / 5 / 5 | 3.5 / 4 / 5 / 5 / 5 |
| **Contextual appropriateness** | $3.82 \pm 1.06$ | $3.43 \pm 1.13$ | $3.42 \pm 1.35$ |
| Median | 4.00 | 3.00 | 3.00 |
| Percentiles (25/50/75/90/95) | 3 / 4 / 5 / 5 / 5 | 3 / 3 / 4 / 5 / 5 | 2.5 / 3 / 5 / 5 / 5 |
| **Error Rates** | | | |
| Factual hallucination | 8.3% | 5.3% | 7.5% |
| Information revelation failures | 4.2% | 5.3% | 5.3% |
| Constraint checking failures | 0.0% | 1.3% | 0.0% |

Table 12: Human evaluation of user simulator quality. Results breakdown by design axis (2) Trust and communication patterns (Tab. 9).

| Metric | Suspicious (n=98) | Trusting (n=68) |
|---|---|---|
| **Quality Scores (1–5) ↑** | | |
| **Clarity** | $3.94 \pm 1.00$ | $3.85 \pm 1.29$ |
| Median | 4.00 | 4.00 |
| Percentiles (25/50/75/90/95) | 3.25 / 4 / 5 / 5 / 5 | 3 / 4 / 5 / 5 / 5 |
| **Contextual appropriateness** | $3.48 \pm 1.15$ | $3.76 \pm 1.11$ |
| Median | 4.00 | 4.00 |
| Percentiles (25/50/75/90/95) | 3 / 4 / 4 / 5 / 5 | 3 / 4 / 5 / 5 / 5 |
| **Error Rates** | | |
| Factual hallucination | 4.1% | 6.8% |
| Information revelation failures | 6.1% | 2.9% |
| Constraint checking failures | 1.0% | 0.0% |

Table 13: Human evaluation of user simulator quality. Results breakdown by design axis (3) Constraint checking reliability (Tab. 9).

| Metric | Reliable (n=124) | Unreliable (n=42) |
|---|---|---|
| **Quality Scores (1–5) ↑** | | |
| **Clarity** | $3.83 \pm 1.17$ | $4.12 \pm 0.96$ |
| Median | 4.00 | 4.00 |
| Percentiles (25/50/75/90/95) | 3 / 4 / 5 / 5 / 5 | 4 / 4 / 5 / 5 / 5 |
| **Contextual appropriateness** | $3.59 \pm 1.12$ | $3.62 \pm 1.21$ |
| Median | 4.00 | 4.00 |
| Percentiles (25/50/75/90/95) | 3 / 4 / 5 / 5 / 5 | 3 / 4 / 5 / 5 / 5 |
| **Error Rates** | | |
| Factual hallucination | 7.1% | 4.8% |
| Information revelation failures | 5.6% | 2.4% |
| Constraint checking failures | 0.8% | 0.0% |

