# OpenReview forum: "COMPASS: A Multi-Turn Benchmark for Tool-Mediated Planning & Preference Optimization"
_ICLR.cc/2026/Conference — ICLR 2026 Conference Withdrawn Submission_

### Official Review · Reviewer_9Th3 · 2025-10-31

**Soundness:** 3
**Presentation:** 3
**Contribution:** 3
**Rating:** 8
**Confidence:** 4

**Summary:**

This paper propose the Compass benchmark that focus on multi-turn preference optimization with rich and complex environments and tools.
They adopt hard constraints and soft preferences and the latter is expected to be optimized through multi-turn interactions.
They also design different levels of tasks to evaluate model performance.
The tools included are diverse and comprehensive, mirroring the real-world environment.
The results show that despite some models can perform well on acceptable rate, they fall short in optimal rate.

**Strengths:**

1. This paper provides a complex environment that mirrors real-world and messy noises rather than a cleaned and simplified simulation.
2. It focuses on the optimal solution rather than an acceptable solution that evaluates how models can best serve as an applicable agent.
3. The multi-turn interactions with the simulated user brings more complications into this problem.

**Weaknesses:**

1. The simulated user query could be deployed to attain more diverse interactions beyond the 241 tasks.

**Questions:**

1. The paper find that agents have higher acceptable rate than the optimal rate. Do you have a hypothesis for why this happens? Is it a failure of insufficient working memory, lack of exploration or sticking to the original solution without searching further?
2. Can the user simulators be deployed to create more diverse and complex test cases?

---

> ### Author Response · Authors · 2025-11-20
> **Response to Reviewer 9Th3**
>
> We thank reviewer 9Th3 for their thoughtful comments! We are glad that you found our paper interesting! We address specific questions and comments below.
>
> > The simulated user query could be deployed to attain more diverse interactions beyond the 241 tasks.
>
> Thank you for this suggestion. Our simulator already supports diverse interactions through a large set of user personas. As described in Section 3.4, we define personas along four behavioral axes (including adversarial and underspecified behaviors), resulting in 108 persona types. These personas are randomly paired with 241 tasks, ensuring extensive interaction diversity.
>
> __We have added summary of 108 user types in in Appendix E.1 and additional analyses in Appendix A.3 (Table 5, Fig. 6)__. These show that strong models perform consistently across user types, while weaker models (including some frontier closed-source models) exhibit large variability. Together, these results demonstrate that our benchmark already provides a wide spectrum of user–agent interaction dynamics.
>
>
> > The paper finds that agents have a higher acceptable rate than the optimal rate. Do you have a hypothesis for why this happens? Is it a failure of insufficient working memory, lack of exploration or sticking to the original solution without searching further?
>
> Great question. As discussed in Section 6.2, our case study suggests that the acceptable–optimal gap arises from models’ limited ability to systematically explore the solution space and reason about whether further search is needed to find a better solution.
> __Additionally, we also conducted a more extensive tool call analysis (Section  6.1)__. The insights below might shed more light on why models fail to systematically perform optimization.
>
> We summarize key insights below:
>
> 1. __Correctness matters__: Table 3 shows clear quality gaps. Closed-source models make almost no tool errors (<1%), while open-source models frequently hallucinate parameters, exposing a gap between open and closed source models.
>
> 2. __Reasoning depth differs__: The tool-call extensiveness study (Fig. 4) reveals that stronger models (e.g., GPT-5) perform deeper, more effective searches by making more tool calls without performance drop.  Weaker models (even frontier models like GPT-4o) degrade as tool calls increase, exposing reasoning limits under complex retrieval. Importantly, open-source models do not even have the capability to invoke many tool calls, which fundamentally limits their search ability.
>
> 3. __Efficiency signals failure__: The tool-call efficiency analysis (Fig. 4) finds that even some frontier models make excessive duplicate calls and this behavior strongly correlates with poor task performances, showing that inefficient reasoning loops are a key failure mode in multi-turn tool use.
>
> Together, these results suggest that the gap is driven primarily by insufficient exploration and weak search efficiency. This analysis also provides actionable insights for post-training teams to address key failure modes in constrained optimization.
>
> > Can the user simulators be deployed to create more diverse and complex test cases?
>
> Thank you for this suggestion. Our current simulator already supports diverse and complex interactions through a wide range of user personas. As described in Section 3.4, we define personas along four behavioral axes (including adversarial and underspecified behaviors), resulting in 108 persona types. These are randomly paired with 241 tasks, ensuring large-scale interaction diversity. Summary statistics are provided in Appendix E.1.
>
> Importantly, our simulator is highly modular, allowing easy extension to new behavioral axes or interaction types. Future work can thus build on our design to create even more diverse and complex user scenarios tailored to specific deployment needs.

---

> > ### Author Response · Authors · 2025-11-26
> >
> > Thank you reviewer 9Th3 again for your time and for the detailed feedback you provided on our submission. We wanted to check in to see whether you have any follow-up questions or remaining concerns that we can clarify.

---

### Official Review · Reviewer_Raxv · 2025-11-01

**Soundness:** 2
**Presentation:** 2
**Contribution:** 1
**Rating:** 2
**Confidence:** 5

**Summary:**

The paper introduces COMPASS, a benchmark for evaluating LLM agents’ constrained preference optimization in travel planning, addressing gaps in existing tool-use and planning benchmarks. However, critical limitations in novelty prevent it from meeting ICLR’s standards for publication.

**Strengths:**

1. The benchmark design integrates realistic travel data (20 U.S. National Parks), a multi-turn user simulator, and a comprehensive tool ecosystem, which aligns with real-world agent deployment scenarios.

2. The identification of "acceptable–optimal gap" and "plan-coordination gap" provides insights for future agentic travel planning.

**Weaknesses:**

1. The primary issue is that the core idea of integrating constrained satisfaction with preference optimization is built upon existing research without introducing substantive innovations. For instance, TravelPlanner initially addressed hard constraint satisfaction, while subsequent works such as ChinaTravel [1] have emphasized the synergy between hard and soft constraints, and TripTailor [2] has also explored aspects of personalization. However, this paper entirely overlooks these aspects, thereby significantly undermining its claimed contributions.

2. Although the authors focus on constraint satisfaction problems, the Related Work section predominantly discusses tool-use benchmarks, while largely ignoring established benchmarks in travel planning—a domain that has seen extensive research. This omission represents a notable gap in the literature review.

3. The proposed level classification lacks intuitive justification. Why not use the number of constraints as a basis for level categorization, which would offer a more straightforward and interpretable framework?

4. The concept of an "acceptable–optimal gap" raises questions about its practical relevance. Under hard constraints, single-objective preference optimization admits a deterministic solution. While achieving this optimum may be challenging for models, it is debatable whether reaching the true optimum is necessary in real-world scenarios. What, then, is the substantive significance of this gap?

5. The relationship between the "plan-coordination gap" and constrained optimization remains unclear and appears conceptually disjointed. The connection between the two proposed gaps and how LLMs address constraint satisfaction problems is neither well-motivated nor clearly articulated.

6. The evaluation methodology for hard and soft constraints is inadequately defined. Using the set of feasible solutions as a metric is problematic: when the number of constraints is small and the database is large, the feasible set can become excessively large—or even impossible to enumerate—rendering evaluation infeasible. While such a measure may be meaningful for preference evaluation, insisting on optimality for all preferences is arguably unreasonable. This touches upon multi-objective optimization (MOO), where Pareto optimality should be considered. Why not employ model-based ranking for preference evaluation instead?


7. The experiment completely ignored Agent-based algorithms, such as ReAct and LLM-module. Why?



[1] ChinaTravel: An Open-Ended Benchmark for Language Agents in Chinese Travel Planning. 2024.

[2]  TripTailor: A Real-World Benchmark for Personalized Travel Planning. 2025.

**Questions:**

see weaknesses

---

> ### Author Response · Authors · 2025-11-21
> **Response to Reviewer Raxv 1/3**
>
> Thank you, Reviewer Raxv, for your thoughtful and constructive feedback. We appreciate the detailed comments and address them below:
>
> Before addressing specific points, we would like to clarify that our benchmark goes __beyond constraint satisfaction__ by framing travel planning as a [__constrained optimization problem__](https://en.wikipedia.org/wiki/Constrained_optimization). As updated in Section 3.1, this formulation explicitly defines the task as:
>
> $$\max_x f(x) \quad \text{s.t.} \quad g_i(x) = c_i, \, h_j(x) \ge d_j$$
>
> Here, users’ hard requirements correspond to constraints $g_i$, $h_j$, while soft preferences define the objective function $f(x)$. The LLM must therefore find a solution that __satisfies constraints__ while also __optimizing user preferences__. The objective $f(x)$ captures both (1) single-objective goals (e.g., “cheapest hotel”) and (2) multi-objective trade-offs where Pareto optimality arises.
>
>
> We address specific comments below.
>
>
> > The primary issue is that the core idea of integrating constrained satisfaction with preference optimization is built upon existing research without introducing substantive innovations. For instance, TravelPlanner initially addressed hard constraint satisfaction, while subsequent works such as ChinaTravel [1] have emphasized the synergy between hard and soft constraints, and TripTailor [2] has also explored aspects of personalization. However, this paper entirely overlooks these aspects, thereby significantly undermining its claimed contributions.
>
>
> Thank you for pointing out these relevant works. To our knowledge, TripTailor was first released on arXiv on Aug 1, 2025, after the ICLR submission deadline (Jul 24, 2025), and thus qualifies as concurrent work under the ICLR guidelines. __We have added it to our related work to make the literature review more comprehensive__.
>
> Our benchmark differs in two key ways:
>
>
> 1. __True multi-turn interaction__:
>
>     ChinaTravel and TripTailor are single-turn benchmarks, where the user provides all task details upfront. Although the agent may make multi-step tool calls, there is no back-and-forth conversation with the user. Consequently, these benchmarks do not capture diverse user behaviors such as ambiguity, inconsistency, feedback, or impatience. In contrast, our __user simulator__ (Sec 3.4, App. E) models a wide range of realistic user behaviors, creating a genuinely conversational planning setting. In the revised version, __we added a user-type analysis (Appendix A.3)__, showing that strong models maintain consistent performance across diverse communication styles, while weaker or even frontier models show large variability. This finding highlights that our multi-turn setup reveals nuanced model behaviors not captured by prior benchmarks.
>
>
>  2. __Systematic evaluation of preference optimization__:
>
>     While both ChinaTravel and TripTailor mention personalization, neither evaluates the agent’s ability to __optimize user preferences__. ChinaTravel primarily follows TravelPlanner’s focus on constraint satisfaction, and TripTailor centers on progressive constraint revelation without modeling soft preferences. Our work is the __first systematic attempt to evaluate constrained preference optimization__, an essential yet previously unexplored dimension in this domain (as also noted by Reviewer 3qtC).
>
>     __In the revised paper, we further strengthen this contribution by adding a new tool-call analysis section (Sec. 6.1)__. This analysis shows that systematic evaluation of preference optimization naturally exposes how models leverage tools to explore and refine the solution space. Specifically, we find that stronger models (e.g., GPT-5) perform deeper, more efficient tool-based searches, whereas weaker models exhibit redundant or shallow tool use. This addition provides a more mechanistic understanding of why current models still struggle with effective optimization in realistic planning tasks.
>
>
> > Although the authors focus on constraint satisfaction problems, the Related Work section predominantly discusses tool-use benchmarks, while largely ignoring established benchmarks in travel planning—a domain that has seen extensive research. This omission represents a notable gap in the literature review.
>
> The extensive literature review on planning was originally included in Appendix B. __We have moved up to the main paper according to your suggestions__. Please let us know if we are missing any work you are aware of.

---

> ### Author Response · Authors · 2025-11-21
> **Response to Reviewer Raxv 2/3**
>
> >  The concept of an "acceptable–optimal gap" raises questions about its practical relevance. Under hard constraints, single-objective preference optimization admits a deterministic solution. While achieving this optimum may be challenging for models, it is debatable whether reaching the true optimum is necessary in real-world scenarios. What, then, is the substantive significance of this gap?
>
> We appreciate this thoughtful question. The __acceptable–optimal gap__ captures an agent’s ability to not only satisfy hard constraints but also optimize user preferences—an essential component of user satisfaction in real-world planning tasks.
>
> For example, consider a user asking the LLM to “book flights for a family of four” with hard constraints (destination, dates) and a soft preference (“find the cheapest option”). From a user perspective, it is natural to expect that the agent’s choice is near-optimal—not just feasible. The acceptable–optimal gap directly measures whether the agent’s solution achieves this standard.
>
> To make this practical, we __relax optimality__ by reporting whether the solution is within the *top 5%, 10%, or 20%* of all valid options. Even under this relaxed criterion, frontier models frequently settle for suboptimal solutions. This demonstrates that current LLMs still lack reliable __optimization capabilities__, a limitation highly relevant to real-world deployment.
>
> > The relationship between the "plan-coordination gap" and constrained optimization remains unclear and appears conceptually disjointed. The connection between the two proposed gaps and how LLMs address constraint satisfaction problems is neither well-motivated nor clearly articulated.
>
> Thank you for the question. Both gaps are closely connected to our __constrained optimization__ formulation.
>
>   - __Acceptable–optimal gap__: This gap measures the difference between constraint satisfaction and preference optimization. The __acceptable rate__ corresponds to plans that satisfy all constraints $(g_i, h_j)$, while the __optimal rate__ reflects plans that also maximize the objective function $f(x)$.
>
>
>   - __Plan–coordination gap__: This gap captures how model performance declines as the constrained optimization problem becomes more complex—i.e., when tasks require coordinating multiple interdependent components (e.g., flights, hotels, attractions). As the number and coupling of constraints increase, both constraint satisfaction and optimization become harder, revealing models’ scalability limits in solving realistic constrained optimization problems.
>
> > The evaluation methodology for hard and soft constraints is inadequately defined. Using the set of feasible solutions as a metric is problematic: when the number of constraints is small and the database is large, the feasible set can become excessively large—or even impossible to enumerate—rendering evaluation infeasible. While such a measure may be meaningful for preference evaluation, insisting on optimality for all preferences is arguably unreasonable. This touches upon multi-objective optimization (MOO), where Pareto optimality should be considered. Why not employ model-based ranking for preference evaluation instead?
>
> We define evaluation methodology in Section 3.5. __We have updated the definition now with the more mathematical definition__. Please let us know if this clarifies things.
>
>  __Acceptable rate__ checks feasibility: does x' satisfy all hard constraints $\lbrace g_i(x')=c_i,\; h_j(x')\ge d_j\rbrace$
>
>  __Optimal rate__ measures preference optimality: how close $f(x')$ is to the ground-truth optimum $f(x^\*)$.
>
>
> __Feasible-set tractability__: Our dataset is built from curated databases. Within this controlled domain, we __enumerate all feasible plans__ and compute $f(x)$ exactly to obtain $x^\*$. This yields a __deterministic, reproducible__ target for optimization—no heuristic search or model judges are required.
>
>
> __Why not model-based ranking?__ LLM judges introduce bias, variance, and can be unreliable. When the user’s objective $f(\cdot)$ is explicit (e.g., price), comparing $f(x')$ to $f(x^\*)$ is __more rigorous and reliable__ than asking another LLM to rank outputs.
>
>
> __Practical relevance__: Even under relaxed optimality, frontier models often produce suboptimal $x'$. Our new tool-call analysis (Sec. 6.1) further shows that this acceptable–optimal gap stems from inefficient or redundant tool use and shallow search behaviors. This highlights that models not only fail to optimize user preferences but also lack the structured reasoning and tool-use strategies necessary for real-world optimization tasks.

---

> ### Author Response · Authors · 2025-11-21
> **Response to Reviewer Raxv 3/3**
>
> > The proposed level classification lacks intuitive justification. Why not use the number of constraints as a basis for level categorization, which would offer a more straightforward and interpretable framework?
>
> Thank you for the thoughtful suggestion. Our current level classification is based on __user request types__—for example, (1) booking a single flight vs. (2) planning a full itinerary. As the level increases, the LLM must coordinate across more travel components, making the underlying optimization problem more complex.
> Since our benchmark focuses on __preference optimization__ rather than pure constraint satisfaction, this design better captures the coordination and reasoning complexity of user goals. That said, we agree that incorporating the number of constraints offers a complementary and interpretable dimension. __We have updated the text (Line 205) to explicitly include constraint count as an additional categorization criterion, making the level taxonomy richer and more meaningful__.
>
> > The experiment completely ignored Agent-based algorithms, such as ReAct and LLM-module. Why?
>
> Thank you for the suggestion. __We have now included ReAct results (GPT models for now, and we are running the rest) in Appendix A.4__. Adding agent-based algorithms makes our benchmark evaluations more complete!
>
> We note that many frontier models (e.g., GPT-5, Claude Opus, Gemini pro) already include built-in reasoning modes, so traditional ReAct-style prompting might not be most suitable. We are currently working on incorporating more modern agentic workflows, such as Reflexion [1], which enables self-evaluation, and LangChain-style architectures that better manage memory and state tracking across multi-turn interactions. LangChain-style agentic flow with better memory tracking is particularly relevant because it will allow agents to maintain evolving user preferences and handle large volumes of tool-call outputs more effectively—capabilities that are critical for complex, conversational planning tasks.
>
>
> [1] Reflexion: Language Agents with Verbal Reinforcement Learning (https://arxiv.org/abs/2303.11366)

---

> > ### Author Response · Authors · 2025-11-26
> >
> > Thank you reviewer Raxv again for your time and for the detailed feedback you provided on our submission. We wanted to check in to see whether you have any follow-up questions or remaining concerns that we can clarify.

---

### Official Review · Reviewer_3qtC · 2025-11-01

**Soundness:** 3
**Presentation:** 2
**Contribution:** 3
**Rating:** 4
**Confidence:** 3

**Summary:**

- Proposes COMPASS, a multi-turn benchmark for tool-mediated travel planning and preference optimization.

- Builds an interactive evaluation framework with tools, databases, and a user simulator reflecting dynamic user behaviors.

- Evaluates several LLM agents, revealing gaps in preference optimization and multi-tool coordination performance.

**Strengths:**

I find this paper’s contribution significant in its dataset and evaluation framework.

- **Dataset**
    - The dataset itself is highly valuable. While most prior works rely on synthetic data, this paper collects a *real-world* dataset through APIs, which makes it much more realistic and potentially very useful for other researchers in this domain.
- **Systematized Evaluation Framework**
    - The authors integrate various evaluation aspects from prior travel-planning studies into a systemized framework.
    - To my knowledge, this is the first systematic attempt to evaluate preference optimization, which is an essential dimension in this domain.
    - **Answer construction**: The authors consider all possible combinations of factors when constructing reference answers, and then define metrics accordingly. This goes beyond simple “pass/fail” checks for each constraint and represents a thoughtful attempt to capture composite preferences.

**Weaknesses:**

While the dataset and framework contributions are strong, some parts, especially the experimental section, leave room for improvement.

**w1. Dataset description insufficiency**

  * The paper lacks a detailed explanation of how the dataset was collected via APIs.
  * While collecting *real-world* data is commendable, the authors should provide a short case study illustrating **what aspects of real-world interactions are captured that synthetic datasets fail to model**.

 **w2. Missing citations / related work**

  * There exist prior works on **multi-turn travel planning** (e.g., [1]) that address similar interactive planning settings, which is similar to one of their proposed user settings (*progressive constraint revelation*).

 **w3. Framework design clarity**

  * *User Simulator Design* (around L250) introduces within-conversation dynamics and persona diversity, which are interesting ideas.
  * However, it remains unclear **whether these behaviors are grounded in the real-world dataset** or are purely heuristic / author-defined.
  * The rationale and distribution (e.g., “how often users exhibit such behaviors”) should be explicitly stated.

 **w4. Framework validation**
  * While authors provide validation of framework in Table 2, It would be helpful if the validation were reported separately for each simulation type.

 **w5. Limited evaluation depth**

  * The results are not sufficiently analyzed across different **user simulation types**, though model behavior likely varies under each setting.
  * Although tool-calling is presented as a key feature of the benchmark, the paper does not analyze **how often models succeed or fail at tool calls**, nor how such success/failure impacts final outcomes.
  * (Minor) The discussion in L411–413 about open-source model tendencies seems weak, since only the Qwen models were evaluated, limiting generalizability.

**w6. Limited novelty/insight in findings**

  * The contribution of collecting a real-world dataset is valuable, but some findings derived from it struggle to present clear takeaways beyond the dataset itself.
  * For instance:

    * Section 5.1 on *constraint conflicts* somewhat overlaps with the findings of [1].
    * Section 5.2 on *conversation efficiency* reproduces patterns already observed in prior reasoning tasks ([2]).

**w7. Presentation and display issues (minor)**

  * Tables and figures are placed far from where they are discussed, disrupting readability (e.g., Figure 2 on page 2).
---
Reference

[1] Flex-TravelPlanner: A Benchmark for Flexible Planning with Language Agents, Oh et al, 2025

[2] MINT: Evaluating LLMs in Multi-turn Interaction with Tools and Language Feedback, Wang et al, 2023

**Questions:**

Already covered in Weaknesses.

---

> ### Author Response · Authors · 2025-11-20
> **Response to Reviewer 3qtC (W1-W3)**
>
> We thank reviewer 3qtC for their thoughtful comments! We address specific questions below.
>
> > w1. Dataset description insufficiency. The paper lacks a detailed explanation of how the dataset was collected via APIs.
> While collecting real-world data is commendable, the authors should provide a short case study illustrating what aspects of real-world interactions are captured that synthetic datasets fail to model.
>
>
> Thank you for this helpful suggestion. We agree that data realism is a key contribution. __We have added Appendix F.1 to describe how the dataset was collected via real-world APIs, and Appendix F.2 to include short case studies illustrating nuances that synthetic datasets often miss.__
>
> We provide two interesting examples:
>
>   1. __Price variation patterns:__ As shown in Fig. 16, accommodation prices exhibit both weekly surges (e.g., weekends) and long-term trends (e.g., seasonal shifts). While weekly patterns anti-correlate with room availability, long-term patterns do not. A truly capable agent could discover and leverage these trends to guide its search toward better or more cost-efficient itineraries.
>
>
>   2. __Location variation patterns:__ Certain destinations systematically offer more high-end options than others. Thus, when planning for luxury travelers, an agent must recognize that some destinations may inherently underperform in matching high-end preferences.
>
>
>
> > w2. Missing citations / related work. There exist prior works on multi-turn travel planning (e.g., [1]) that address similar interactive planning settings, which is similar to one of their proposed user settings (progressive constraint revelation).
>
> Thank you for pointing out this missing citation—__we have added it to the related work section__. While [1] extends TravelPlanner by introducing progressive constraint revelation, it only evaluates whether models can adapt plans as users reveal new constraints. In contrast, our work proposes a systematic constrained __preference optimization framework__ built on a realistic, API-based database.
>
> Crucially, our benchmark evaluates an agent’s ability to optimize among feasible solutions to best satisfy user preferences—an aspect neither [1] nor TravelPlanner [2] considers. We have clarified this distinction in Section 3.1 to better highlight our novel contribution.
>
>
> > w3. Framework design clarity. User Simulator Design (around L250) introduces within-conversation dynamics and persona diversity, which are interesting ideas. However, it remains unclear whether these behaviors are grounded in the real-world dataset or are purely heuristic / author-defined.
>
> Thank you for the question. The within-conversation dynamics in our simulator are grounded in real-world dialogue data, not purely heuristic. Our design builds on established work in task-oriented dialogues [3–6], where human annotators systematically studied and categorized user behaviors such as ambiguity, inconsistency, and feedback in settings like travel planning.
>
> While exact behavior distributions naturally depend on deployment environments, our simulator is *fully controllable*—allowing practitioners to adjust behavior frequencies based on their own user data. This design ensures both realism and adaptability across application contexts.

---

> ### Author Response · Authors · 2025-11-20
> **Response to Reviewer 3qtC (W4-W7)**
>
> > w4. Framework validation. While authors provide validation of framework in Table 2, It would be helpful if the validation were reported separately for each simulation type.
>
> Thank you for the helpful suggestion. __We have added per-simulation-type validation results in Tables 9–11 (Appendix H.2).__ The results show no significant quality differences across simulation types, supporting the robustness and consistency of our user simulator design.
>
> > w5. Limited evaluation depth. The results are not sufficiently analyzed across different user simulation types, though model behavior likely varies under each setting. Although tool-calling is presented as a key feature of the benchmark, the paper does not analyze how often models succeed or fail at tool calls, nor how such success/failure impacts final outcomes.
>
> We appreciate this valuable suggestion. __In the revised paper, we added two new analyses that strengthen our benchmark’s evaluation depth:__
>
> 1.  Tool calling analysis (Section 6.1). We summarize key insights below:
>
>     *  __Correctness matters:__ Table 3 shows clear quality gaps. Closed-source models make almost no tool errors (<1%), while open-source models frequently hallucinate parameters, exposing a gap between open and closed source models.
>
>
>     *  __Reasoning depth differs:__ The tool-call extensiveness study (Fig. 4) reveals that stronger models (e.g., GPT-5) perform deeper, more effective searches by making more tool calls without performance drop.  Weaker models (even frontier models like GPT-4o) degrade as tool calls increase, exposing reasoning limits under complex retrieval. Importantly, open-source models do not even have the capability to invoke many tool calls, which fundamentally limits their search ability.
>
>
>     *  __Efficiency signals failure:__ The tool-call efficiency analysis (Fig. 4) finds that even some frontier models make excessive duplicate calls and this behavior strongly correlates with poor task performances, showing that inefficient reasoning loops are a key failure mode in multi-turn tool use.
>
>  2. User type analysis (App. A.3) . We also summarize key insights below:
>
>     *  __Information revelation speed matters__. When users reveal task constraints slowly (i.e., across many turns), model performance degrades across both open- and closed-source systems. This shows that delayed constraint specification directly harms preference optimization and constraint satisfaction in realistic, multi-constraint tasks.
>
>     *  __Trust level has limited effect__. Prior work [7] studies feedback handling under oracle-like assumptions where user feedback is always informative. In contrast, our results show that blind suspicion—users prompting the model to “double-check” without adding any new information—does not improve performance for any tested model. This highlights current limits in models’ self-verification abilities.
>
>     *  __Communication style robustness varies by model__. Strong models (e.g., GPT-5) maintain stable performance across diverse communication styles (polite, rude, terse, verbose), whereas weaker and some frontier models remain sensitive to user tone, politeness level, and verbosity. We quantify robustness using the Coefficient of Variation (CV), finding large gaps across model families.
>
>
>
> > w6. Limited novelty/insight in findings. The contribution of collecting a real-world dataset is valuable, but some findings derived from it struggle to present clear takeaways beyond the dataset itself.
> For instance:
> Section 5.1 on constraint conflicts somewhat overlaps with the findings of [1].
> Section 5.2 on conversation efficiency reproduces patterns already observed in prior reasoning tasks ([2]).
>
> We appreciate this comment and have clarified our main contributions accordingly. Our key finding—___the acceptable–optimal gap__ (Fig. 2)—shows that even frontier models (e.g., GPT-5) often settle for solutions that merely satisfy hard constraints without optimizing for user preferences. __While prior works have discussed constraint satisfaction ability [1, 2], none have examined models’ constrained optimization ability in realistic, multi-turn settings.__
>
> Moreover, __our new tool-call analysis (Section 6.1)__ reveals that this gap arises from models’ limited ability to plan, conduct targeted searches via tools, and synthesize complex contextual information. To our knowledge, no previous benchmark has systematically evaluated this optimization capability, which we believe is essential for practical planning agents.
>
>
> > w7. Presentation and display issues (minor)
> Tables and figures are placed far from where they are discussed, disrupting readability (e.g., Figure 2 on page 2).
>
> We have fixed figure placements in the revised version.

---

> > ### Author Response · Authors · 2025-11-20
> > **References**
> >
> > [1] FlexTravel-Planner (https://arxiv.org/pdf/2506.04649)
> >
> > [2] TravelPlanner (https://arxiv.org/abs/2402.01622)
> >
> > [3] GenTUS: Simulating User Behaviour and Language in Task-oriented Dialogues with Generative Transformers (https://arxiv.org/abs/2208.10817)
> >
> > [4] MultiWOZ – a large-scale multi-domain wizard-of-oz dataset for task-oriented dialogue modelling. ( http://arxiv.org/abs/1810.00278.)
> >
> > [5] Towards scalable multi-domain conversational agents: The schema-guided dialogue dataset. (https://arxiv.org/abs/1909.05855)
> >
> > [6] A simple language model for task-oriented dialogue. (https://arxiv.org/abs/2005.00796)
> >
> > [7] MINT: Evaluating LLMs in Multi-turn Interaction with Tools and Language Feedback (https://arxiv.org/abs/2309.10691)

---

> ### Author Response · Authors · 2025-11-25
>
> Thank you reviewer 3qtC again for your time and for the detailed feedback you provided on our submission. We wanted to check in to see whether you have any follow-up questions or remaining concerns that we can clarify.

---

### Official Review · Reviewer_YyYx · 2025-11-01

**Soundness:** 2
**Presentation:** 3
**Contribution:** 2
**Rating:** 4
**Confidence:** 4

**Summary:**

This paper introduces a multi-turn benchmark for evaluating Large Language Model (LLM) agents on tool-mediated planning and constrained preference optimization within a travel planning environment. The benchmark's evaluation reveals that current state-of-the-art LLMs can satisfy basic constraints but consistently fall short in finding truly optimal, preference-aligned solutions, particularly as task complexity increases.

**Strengths:**

1. The paper is original in how it formalizes multi-turn travel planning as constrained preference optimization, explicitly separating hard feasibility constraints from soft preference objectives. This framing enables measurable evaluation of both “acceptable” solutions and optimization quality.
2. The benchmark design is reasonable. It couples a realistic tool ecosystem (hotels, flights, permits) with a dynamic, persona-driven user simulator and provides ground-truth optima via exhaustive enumeration over feasible combinations. Metrics are interpretable and complementary, and the analysis is thorough across plan-coordination levels, constraint count, search complexity, and conversation efficiency.

**Weaknesses:**

1. External validity is constrained by the LLM-based user simulator. Despite some human audits, the simulator’s coverage of adversarial, ambiguous, or inconsistent user behaviors is limited, and there is no direct comparison with real users.
2. The domain scope is narrow, which can even be considered an extension of the TravelPlanner benchmark. Triptailor[1] is also a multi-turn travel planning benchmark, which involves more detailed itineraries and personalized requirements. Treating this work as an interface design and analyzing the challenges of the task on different benchmarks would make a greater contribution to the community.
3. The analysis of tool usage is crucial in travel planning, as discussed in section 5.3. However, it's unfortunate that only a case study is provided there. Is there a better mechanism for quantitative analysis in this area?
4. The paper's title emphasizes "Tool-Mediated Planning and Preference Optimization," yet a substantial portion of the content is dedicated to addressing constraints of varying difficulty levels. This creates a certain degree of misalignment between the stated focus and the actual technical emphasis.

[1] Triptailor: A real-world benchmark for personalized travel planning

**Questions:**

1. Simulator coverage: How diverse are personas and scripts relative to the full task space? Can you provide coverage analyses and persona fidelity checks, including adversarial or underspecified behaviors?
2. In an agent that uses multiple rounds of tool calls to ultimately achieve planning, how should the agent consider calling the tools, how should the tools be called, and whether the call is successful? What methods are available for analyzing the capabilities of an LLM in this regard?

---

> ### Author Response · Authors · 2025-11-20
> **Response to reviewer YyYx's comments**
>
> We want to thank reviewer YyYx for their thoughtful comments! We address specific questions below.
>
> > External validity is constrained by the LLM-based user simulator. Despite some human audits, the simulator’s coverage of adversarial, ambiguous, or inconsistent user behaviors is limited, and there is no direct comparison with real users.
>
> A diverse user simulator design is central to our benchmark. Our user simulator explicitly models both ambiguous and inconsistent behaviors, as detailed in Section 3.4 and Appendix E.1. We do not directly compare with real users because our simulator builds on well-established frameworks for task-oriented dialogue [1–4], which have already characterized such user behaviors.
>
> > The domain scope is narrow, which can even be considered an extension of the TravelPlanner benchmark. Triptailor[1] is also a multi-turn travel planning benchmark, which involves more detailed itineraries and personalized requirements.
>
> Thank you so much for pointing out a highly relevant work! We appreciate the pointer to TripTailor [1]. To our knowledge, it was first released on arXiv on Aug 1, 2025, after the ICLR submission deadline (Jul 24, 2025), and thus qualifies as concurrent work under the official ICLR guidelines.
>
> That said, __we have added to our related work section!__. Our benchmark differs in two substantial ways:
>
> 1. __Optimization-based evaluation__: We introduce a constrained optimization framework to assess whether an agent can identify the best itinerary under user-defined constraints and preferences. Neither TravelPlanner nor TripTailor uses this formulation. While TripTailor considers personalization, it does not evaluate optimization across multiple feasible plans. Our framework explicitly measures whether the agent can maximize user satisfaction—for example, whether it can guarantee that “95% of the time, the agent finds the best option according to your needs.” This notion of success rate directly reflects real-world reliability.
>
>
> 2.  __True multi-turn user interaction__:  Our benchmark uniquely models multi-turn conversations where users may be ambiguous, inconsistent, or provide feedback and corrections. In contrast, TravelPlanner and TripTailor are effectively single-turn—users provide all task details upfront, without iterative exchange. By incorporating dynamic user behaviors such as clarification or impatience, our benchmark better captures the challenges faced by real conversational agents in practical deployment.
>
>
> > The analysis of tool usage is crucial in travel planning, as discussed in section 5.3. However, it's unfortunate that only a case study is provided there. Is there a better mechanism for quantitative analysis in this area?
>
> We appreciate this valuable suggestion. We fully agree that quantitative analysis of tool use is critical and makes the paper much stronger. __In the revised PDF, we used the additional page to provide a detailed quantitative study (Sec 6.1).__ Key findings include:
>
> 1. **Correctness matters**:Tab. 3 shows clear quality gaps. Closed-source models make almost no tool errors (<1%), while open-source models frequently hallucinate parameters, exposing a gap between open and closed source models.
>
> 2. __Reasoning depth differs__ : The tool-call extensiveness study (Fig. 4) reveals that stronger models (e.g., GPT-5) perform deeper, more effective searches by making more tool calls without performance drop.  Weaker models (even frontier models like GPT-4o) degrade as tool calls increase, exposing reasoning limits under complex retrieval. Importantly, open-source models do not even have the capability to invoke many tool calls, which fundamentally limits their optimization ability.
>
> 3. __Efficiency signals failure__ :The tool-call efficiency analysis (Fig. 4) finds that even some frontier models make excessive duplicate calls and this behavior strongly correlates with poor task performances, showing that inefficient reasoning loops are a key failure mode in multi-turn tool use.
>
> > The paper's title emphasizes "Tool-Mediated Planning and Preference Optimization," yet a substantial portion of the content is dedicated to addressing constraints of varying difficulty levels.
>
> We appreciate this observation. Our benchmark is designed to evaluate __constrained preference optimization__—that is, an agent’s ability to satisfy hard constraints while optimizing for user preferences. __We have clarified this in Section 3.1, where Eq. (1–3) formally defines the task as a constrained optimization problem.__
>
> Our key finding—__the acceptable–optimal gap__ (Sec. 4.1)—shows that even frontier models often stop at satisfying constraints without further optimizing for user preferences. In Section 5.1, we analyze both constraint complexity and optimization difficulty, as both are central to understanding agent performance. We have revised the text to consistently emphasize this framing and to better align the title with the paper’s core technical focus.

---

> > ### Author Response · Authors · 2025-11-20
> > **References**
> >
> > [1] GenTUS: Simulating User Behaviour and Language in Task-oriented Dialogues with Generative Transformers (https://arxiv.org/abs/2208.10817)
> >
> > [2] MultiWOZ – a large-scale multi-domain wizard-of-oz dataset for task-oriented dialogue modeling.(http://arxiv.org/abs/1810.00278.)
> >
> > [3] Towards scalable multi-domain conversational agents: The schema-guided dialogue dataset. (https://arxiv.org/abs/1909.05855)
> >
> > [4] A simple language model for task-oriented dialogue. (https://arxiv.org/abs/2005.00796)

---

> > ### Author Response · Authors · 2025-11-20
> > **Response to reviewer YyYx's questions**
> >
> > >  Simulator coverage: How diverse are personas and scripts relative to the full task space? Can you provide coverage analyses and persona fidelity checks, including adversarial or underspecified behaviors?
> >
> > Thank you for this suggestion. Our simulator incorporates a diverse set of user personas.  As detailed in Section 3.4, we define personas along four axes (including adversarial and underspecified behaviors), yielding _108_ persona types. These are randomly paired with 241 tasks to ensure broad coverage across the task space.
> >
> >  __We have added overall diversity statistics in Appendix E.1, and additional analyses in Appendix A.3 (Table 5, Fig. 6)__ . These show that while strong models perform consistently across user types, weaker ones (including some frontier closed-source models) exhibit significant variability—highlighting both the fidelity and discriminative power of our simulator design.
> >
> > > In an agent that uses multiple rounds of tool calls to ultimately achieve planning, how should the agent consider calling the tools, how should the tools be called, and whether the call is successful? What methods are available for analyzing the capabilities of an LLM in this regard?
> >
> > Please see our detailed response on tool-call analysis (Section 6.1). In our setup, agents are expected to iteratively call multiple tools with varying inputs to explore the solution space and identify the optimal plan. A successful tool-calling process therefore involves making diverse, non-redundant calls and efficiently adjusting parameters to improve search quality. Our __new quantitative analyses__ in Section 6.1 evaluate these behaviors through correctness, extensiveness, and efficiency metrics.

---

> ### Comment · Reviewer_YyYx · 2025-11-23
>
> Thank you for your reply, and for the extensive explanations and experiments you provided. Some concerns have been addressed, while others remain, as detailed below.
> **W1.  LLM-based user simulator.**.
> The diversity concern has been addressed, but how to ensure quality is still not clear.
> Thank you for the detailed clarification which made the user simulation more clear. Table 9 shows the changes in the simulated user's LLM prompt across four dimensions: Progressive Constraint Revelation, Trust and Communication Patterns, Constraint-Checking Reliability, and Conversation Style. How do these different user types affect preference optimization? Has the author analyzed this aspect? If so, it would be helpful.
>
> **W2. Benchmark Extension**.
>
> **The ICLR submission deadline is Sep. 24, 2025, not Jul. 24, 2025.**
>
> **W3. Tool-use analysis**.
> Thank you for providing the detailed analysis, comparing whether LLMs used tools correctly and the frequency of tool usage. This undoubtedly makes this work more solid. What I'd like to know is, besides analyzing the preprocessing checks such as whether the tools and parameters selected by the LLMs during tool calls are correct, can tool calls be linked to the overall task objective, i.e., preference optimization? How does the information retrieved by tool calls affect the final constraint satisfaction and preference optimization?

---

> ### Author Response · Authors · 2025-11-25
> **Response to Reviewer YyYx**
>
> Thank you for your time in engaging in the conversation. Your questions and comments have been helping us strengthened our paper. We provide response below.
>
> __W1. LLM-based user simulator__
>
> To ensure quality of the simulator, we used an independent third-party professional annotator to validate the user simulator. The main human evaluation results are presented in Sec. 4.2 (Tab. 2), and we additionally report evaluation by user type in App. H.2 (Tab. 11–13).
>
> Regarding your question “How do these different user types affect preference optimization?”:
> In our most recent revision, we added a dedicated analysis section examining model performance across user types. We summarize the key findings below:
>
>    - __Information revelation speed__. When users reveal task information slowly, performance degrades across both open- and closed-source models. This shows that delayed task specification directly harms preference optimization (App. A.3; Fig. 5).
>
>    - __Trust levels__. Blind suspicion, i.e., users asking the model to “double-check” its answer without providing new information or specific feedbacks, does not improve performance for any model we tested. This indicates that current LLMs do not reliably benefit from self-verification prompts when asked by user (App. A.3; Tab. 6).
>
>
>    - __Communication style__. Strong models (e.g., GPT-5) show stable preference-optimization performance across communication styles (polite, rude, terse, verbose). In contrast, weaker and some frontier models remain sensitive to tone and verbosity. We quantify consistency using the Coefficient of Variation (CV) (App. A.3; Tab. 7, Fig. 6).
>
>
>
> __W2. Benchmark Extension.__
>
> Sorry for the earlier typo. The __July 24, 2025__ date is the cut-off after which works should be considered concurrent. According to the ICLR official reviewer guide (https://iclr.cc/Conferences/2026/ReviewerGuide), any work published in a peer-reviewed venue within __two months__ of the ICLR submission deadline should be treated as concurrent, and authors are not required to compare against it. This is how we derived the July 24 date.
>
>
> __W3. Tool-use analysis.__
>
> We are glad that you think the new tool call analysis has made the work more solid. We now answer your question “Can tool calls be linked to task objective? How does the information retrieved by tool calls affect the final constraint satisfaction and preference optimization”:
>
> This is an insightful and important question. While our earlier analysis evaluated whether models call tools correctly and how frequently they do so, these metrics alone do not reveal whether tool use results in retrieving the right information needed to satisfy constraints and optimize user preferences. For instance, APIs such as search_hotels or search_flights frequently return a large space of possible offers; succeeding requires not only issuing correct tool calls but also collecting sufficient, relevant information and integrating it effectively. To directly study this connection, __we have added a new analysis in Appendix A.5 that examines the information retrieved and its relation to task success__. This complements our tool call analysis in Sec 6.1. We summarize the key findings below:
>
>   1. __Agents that discover more unique bookable offers (hotels, flights, permits) are substantially more likely to produce Top-5% optimal solutions.__ Using unique offer IDs (a feature in our database), we quantify coverage of the available solution space and find that higher coverage strongly correlates with success.
>
>
>   2. __Retrieval of auxiliary informational details (e.g., airport transfers, driving distances, airline pet policies) also improves task success.__ These details require additional tool calls beyond the basic search endpoints. Successful models tend to acquire more such contextual information.
>
>
>   3. __The poorer performance of weaker open-source models is strongly explained by insufficient information gathering rather than inability to reason over retrieved items.__ For example, for Qwen3-8B, exploration (not downstream reasoning) is the primary bottleneck.
>
>
>   4.  __Looking forward, information gathering efficiency is an important open challenge.__ In our current results, models are still in a regime where “more information leads to higher task success.”  However, ideal agents should succeed after gathering only the necessary and high-value information. As we highlight in our case study (Sec. 6.1; Fig. 5), more efficient strategies—such as pruning dominated options early, maintaining a running best itinerary, and reasoning about which searches are likely to be informative—could meaningfully reduce redundant tool calls and lead to more intelligent exploration. We view this as a promising direction for future work.
>
>
> We appreciate the your suggestion—it substantially strengthened our analysis!
>
>
> [1]  MINT: Evaluating LLMs in Multi-turn Interaction with Tools and Language Feedback, Wang et al, 2023

---

> > ### Author Response · Authors · 2025-11-26
> >
> > Thank you reviewer YyYx again for engaging in discussion! We really appreciate your time and thoughtful feedbacks! We wanted to check in to see whether you have any follow-up questions or remaining concerns that we can clarify.

---

### Author Response · Authors · 2025-12-04
**Meta-Response**

Dear Area Chair and Reviewers,

We want to thank you again for your time in this process. We have worked diligently to address all reviewer feedback and believe the revisions have substantially strengthened the paper.

Multiple reviewers recognized our **core contributions**: (1) the first systematic evaluation framework for **constrained preference optimization** in multi-turn settings, (2) a realistic, API-based travel planning dataset, and (3) the identification of the **acceptable–optimal gap** revealing that frontier models satisfy constraints without optimizing user preferences.

### Main Concerns and How We Addressed Them

Reviewers YyYx and 3qtC (both borderline score 4) primarily requested additional analysis and details:

**1. Tool Call Analysis (Both Reviewers):** Added comprehensive Section 6.1 analyzing tool call correctness, extensiveness, and efficiency, plus Appendix A.5 connecting information retrieval to optimization outcomes. Reviewer YyYx confirmed: *"This undoubtedly makes this work more solid"*

**2. User Simulator Validation (Both Reviewers):** Added per-simulation-type validation (Appendix H.2) and user-type performance analysis (Appendix A.3) demonstrating simulator fidelity

**3. Dataset Details (Reviewer 3qtC):** Added Appendix F with collection methodology and case studies illustrating realistic patterns absent in synthetic datasets

### Clarifying Our Core Contribution for Reviewer Raxv

Reviewer Raxv's concerns might stem from a misunderstanding of our paper's focus and we have revised our writing to fix this. Reviewer Raxv frames our work as "constraint satisfaction" and compares it to benchmarks that in that area. However, **our work evaluates constrained preference optimization** (maximizing f(x) subject to constraints), which is **mathematically distinct** from constraint satisfaction. This is formalized in Section 3.1 (Eq. 1-3). Our key finding shows models can achieve good performance on constraint satisfaction but only fails at (preference) optimization. This result demonstrating they stop at feasible solutions without optimizing. Reviewers YyYx and 3qtC explicitly recognized this as *"the first systematic attempt to evaluate preference optimization, which is an essential dimension in this domain."*

Regarding Reviewer Raxv's citation of TripTailor (Aug 1, 2025): this is **concurrent work** per ICLR guidelines (within 2 months of Sep 24 deadline). While TripTailor and ChinaTravel mention personalization, neither provides ground-truth optima or systematically evaluates optimization quality—they focus on constraint satisfaction, not optimization.

### Note on Reviewer Engagement

Reviewer YyYx engaged once during rebuttal, helping strengthen the paper. We provided detailed responses to all reviewers but unfortunately did not receive further feedback from Reviewers 3qtC, Raxv, and 9Th3 despite our requests.

### Summary

We have addressed all feedback with substantial new analyses. Reviewer recognition includes:

- **Reviewer 3qtC:** *"This is the first systematic attempt to evaluate preference optimization, which is an essential dimension"* and praised our *"real-world dataset through APIs, which makes it much more realistic"*
- **Reviewer YyYx:** Confirmed additions *"substantially strengthened"* the work
- **Reviewer 9Th3:** Praised our focus on *"optimal solution rather than acceptable solution"* and *"complex environment that mirrors real-world...noises"*

These comments underscore the novelty of our constrained optimization framework, which evaluates whether agents can truly optimize for user preferences. Optimization is a critical capability for real-world deployment. We believe the revisions comprehensively address all concerns and respectfully request your consideration of the paper's strengthened contributions.

---

### Note · Authors · 2026-04-01

I have read and agree with the venue's withdrawal policy on behalf of myself and my co-authors.

---

### Meta-Review · Area_Chair_gKgP · 2026-01-07

**Summary:**

Generally, the reviewers raised two main concerns of the proposed COMPASS benchmark. First, the task scope is overly limited, as the work is exclusively confined to the travel planning domain. This narrow focus severely restricts the generalizability of the benchmark and its contribution to the broader field of tool-mediated planning and preference optimization. Second, despite the authors’ supplementary analyses during the rebuttal (e.g., tool call analysis, user type performance evaluation), the evaluation depth remains insufficient. Critical gaps include the lack of comprehensive validation across diverse domains. Therefore, given the current version, I think this paper may be below the acceptance bar of ICLR, and I would recommend rejection.

**Reviewer Concerns:**

The authors provided detailed rebuttals with supplementary analyses (e.g., tool call evaluation, user simulator validation, ReAct results), addressing partial technical concerns (e.g., dataset clarity, tool use analysis). Reviewers (e.g., YyYx) noted these strengthened the work, but the two core rejection reasons remain unaddressed.

Generally, regarding the task scope, the authors emphasized the differences from prior work but failed to address the fundamental issue of domain specificity, i.e., focusing exclusively on a single planning domain restricts the benchmark’s generalizability to other planning scenarios. Regarding the research depth,  while supplementary analyses (e.g., user type, tool call) were added, key elements such as cross-domain validation, generalizability of core findings, and comprehensive evaluations of agentic workflows remain largely lacking.

**Reviewer Scores:**

I think all reviewers did not show an explicit and strong willingness to change the score. Thus I think that very likely most of the scores will be maintained.

---

### Decision · Program_Chairs · 2026-01-26

Reject